# A primordial germ cell-like-cell platform enables CRISPRi screen for epigenetic fertility modifiers

Liangdao Li [1,3], Jingyi Gao[1], Dain Yi [1], Alex P Sheft[1], John C Schimenti [1,2✉] & Xinbao Ding [1,3]

## Abstract

**Primordial germ cells (PGCs) are the precursors of gametes, and the ability to derive PGC-like cells (PGCLCs) from pluripotent stem cells has transformed germline research. A key limitation remains producing PGCLCs in sufficient numbers for large-scale applications. Here, we show that overexpression of *Nanog* plus three PGC master regulators - *Prdm1*, *Prdm14*, and *Tfap2c* - in mouse epiblast-like cells and formative embryonic stem cells yields abundant and highly enriched PGCLCs without costly recombinant cytokines. *Nanog* enhances the PGC regulatory network, suppresses somatic differentiation, and stabilizes PGCLC fate. Transcriptomically, these PGCLCs are developmentally more advanced than cytokine-induced counterparts and can be sustained long-term or differentiated into spermatogonia-like cells. Using this platform, we conduct a CRISPRi screen of 701 epigenetic genes to identify those needed for PGCLC formation. Downregulation of *Ncor2*, a histone deacetylase (HDAC) recruiter, has the greatest impact. Additionally, the HDAC inhibitors valproic acid and sodium butyrate suppress PGCLC formation and sperm counts of in utero-exposed animals. This work establishes a scalable system for functional screening of genes that influence germline development.**

**Keywords** PGCLCs; Epigenome; Fertility; CRISPRi; Histone Deacetylase Inhibition
**Subject Categories** Chromatin, Transcription & Genomics; Methods & Resources; Stem Cells & Regenerative Medicine

## Introduction

In mammals, gametogenesis is a complex and long process that is initiated by the specification of primordial germ cells (PGCs), the precursors of oocytes and sperm. Unraveling the mechanisms of PGC development is crucial for understanding key processes such as sexually dimorphic epigenetic inheritance and the bases of infertility, a relatively common health condition impacting ~15% of all couples (Ding and Schimenti, 2021). In mammals, PGCs specified in the embryonic epiblast respond to extrinsic and intrinsic signaling pathways, including bone morphogenetic protein (BMP) and WNT signaling. In mice, BMP4 activates WNT3, and subsequently a master mesodermal transcription factor (TF), brachyury (also known as T). T activates critical the early germline TFs, *Prdm1* (a PR/SET domain-containing protein with Zn fingers, also known as *Blimp1*), *Prdm14* (also a PR/SET domain-containing protein), and *Tfap2c* (a DNA-binding TF also known as *AP2γ*) (Saitou and Hayashi, 2021). *Prdm1* is expressed in the precursors of PGCs, which then stimulates *Prdm14* and *Tfap2c* transcription to activate the germline pathway and repress somatic gene expression (Ohinata et al, 2005; Yamaji et al, 2008; Weber et al, 2010). The core TF network containing *Prdm1*, *Prdm14*, and *Tfap2c* is critical for mouse PGC specification; it represses somatic differentiation while driving reacquisition of a pluripotency network and epigenetic reprogramming (Saitou and Hayashi, 2021). After migration to and colonization of the genital ridges, male germ cells differentiate into gonocytes. Thereafter, they become spermatogonial stem cells (SSCs) that subsequently seed recurrent rounds of mitotic expansion, yielding spermatogonia that enter meiosis, and eventually, haploid spermatozoa.

Characterization of these gene regulatory pathways in PGCs formed the basis for creating mouse primordial germ cell-like cells (PGCLCs) from pluripotent stem cells in vitro (Saitou and Hayashi, 2021). Pluripotency has at least three distinct states that mirror early embryonic development: naive or ground state ESCs resembling the preimplantation epiblast; formative ESCs resembling early post-implantation gastrulating epiblast; and primed epiblast stem cells (EpiSCs) resembling late post-implantation gastrulating epiblast (Nichols and Smith, 2009; Kinoshita et al, 2021; Yu et al, 2021). Naive ESCs typically are cultured with inhibitors of fibroblast growth factor and mitogen-activated protein kinase (FGF/MEK) as well as glycogen synthase kinase 3β (GSK3β) in the presence of leukemia inhibitory factor (LIF; "2i+LIF" medium). EpiSCs self-renew when exposed to Activin A and FGF2. Naive ESCs do not readily respond to germ cell inductive stimuli, while primed EpiSCs display limited germline competence. By modulating activin and WNT signaling, a formative state has been stably captured in vitro (Kinoshita et al, 2021; Yu et al, 2021; Luo et al, 2023; Wang et al, 2021). In this intermediate pluripotency state, formative ESCs possess dual competency for mouse chimera formation (when introduced into host blastocysts) and direct responsiveness to PGC specification.

The ability to model germ cell development in culture holds great potential for applications such as dissecting the genomic regulation of gametogenesis, and identifying genetic causes of

[1]Cornell University, College of Veterinary Medicine, Department of Biomedical Sciences, Ithaca, NY 14853, USA. [2]Cornell University, College of Agriculture and Life Sciences, Department of Molecular Biology and Genetics, Ithaca, NY 14853, USA. [3]These authors contributed equally: Liangdao Li, Xinbao Ding. ✉E-mail: jcs92@cornell.edu

infertility that would be important for assisted reproductive technologies (ART) involving gene corrections and optimization of in vitro gametogenesis (IVG). However, these applications remain challenging due to the limited scalability of PGCLC generation. Conventional methods involve a transient conversion from naive ESCs to germline-competent epiblast-like cells (EpiLCs), which represent a formative state. Aggregated EpiLC clusters are then exposed to high concentrations of BMPs and supporting cytokines, leading to their differentiation into PGCLCs (Hayashi et al, 2011). Alternatively, PGCLC specification can be induced by overexpressing three TFs (*Prdm1*, *Prdm14*, and *Tfac2c*), and to a lesser extent, two TFs (*Prdm1* + *Tfap2c* or *Prdm14* + *Tfap2c*) or even a single TF (*Prdm14*), in EpiLC aggregates (Nakaki et al, 2013). *Nanog*, a core pluripotency factor expressed in PGCs, is not essential for the emergence of PGCs or germline transmission, but its deletion results in a significant reduction in PGC numbers both in vitro and in vivo (Zhang et al, 2018b). *Nanog* is also expressed in naive ESCs, and its expression undergoes a transient downregulation as cells transition into the EpiLC state (Acampora et al, 2017; Hayashi et al, 2011). The overexpression of *Nanog* can promote efficient PGCLC induction by binding endogenous enhancers of *Prdm1* and *Prdm14*, even without BMP4 and associated cytokines (Murakami et al, 2016). Furthermore, Nanog overexpression induces cytokine-free PGCLC specification by repressing *Otx2*, encoding a TF essential for anterior neuroecto-derm specification (Vojtek et al, 2022; Acampora et al, 1995).

Promoting cell proliferation is another strategy for producing a large number of PGCs/PGCLCs. However, an extended culture of mouse PGCs promotes dedifferentiation into pluripotent embryo-nic germ cells (EGCs) (Shamblott et al, 1998; Matsui et al, 1992). Saitou and colleagues found that Rolipram and Forskolin, which stimulate cAMP signaling, enables expansion of PGCLCs up to ~50-fold in the presence of SCF (stem cell factor) on m220 feeders (stromal cells expressing a membrane-bound form of human SCF), and the expanded PGCLCs can contribute to spermatogenesis after transplantation to neonatal testes (Ohta et al, 2017; Ishikura et al, 2021; Murase et al, 2020). Nevertheless, PGCLCs cultured under these conditions cannot be expanded for more than 7 days. Although these methods have been instrumental in understanding and improving PGCLC formation, they are associated with either low efficiency or high costs (for reagents such as cytokines).

While several genetic causes of infertility have been identified (Gajbhiye et al, 2018; Krausz and Riera-Escamilla 2018), pinpoint-ing specific epigenetic factors that perturb gene expression is more challenging. Sperm counts have been reported to have declined worldwide since the mid 1900s—possibly due to environmental factors introduced after the industrial revolution—but there are major concerns regarding data analysis and interpretations under-lying these claims (Jørgensen et al, 2021). Exposure to certain chemicals during fetal development, such as endocrine disruptors, can cause transgenerational alterations in epigenetic landscapes and hinder reproductive function and fertility in both males and females (Brieño-Enríquez et al, 2016; Rattan et al, 2017; Pivonello et al, 2020; Skinner, 2007). During PGC development, there is a global loss of repressive histone marks such as H3K9me2 and an increase of active histone marks such as H3K4me3 and H3K27ac (Kurimoto et al, 2015; Tang et al, 2015). These dynamic epigenetic landscapes are crucial, and exposure to specific environmental toxins or chemicals that interfere with the epigenetic profile may

impact PGC development (Lombó et al, 2019; Li et al, 2019). Histone acetylation and deacetylation regulate gene expression and repression, respectively. Nuclear receptor co-repressor 2 (NCOR2) is a well-studied nuclear receptor co-repressor that facilitates histone deacetylation (Nagy et al, 1997; Huang et al, 2000). There are 18 histone deacetylases (HDACs) that are evolutionarily conserved across all eukaryotes and can be divided into four classes (Park and Kim, 2020). Both Class I (HDAC1, 2, and 3) and Class IIa (HDAC4, 5, 7, and 9) are known to interact with NCOR2 complexes to help downregulate gene expression (Nagy et al, 1997; Huang et al, 2000). Trichostatin A, an inhibitor of Class I and II HDACs (HDACi), promotes PGC dedifferentiation into EGCs, suggesting that histone deacetylation is important for maintaining the PGC state (Durcova-Hills et al, 2008).

Here, we report a simple, scalable, and economical approach to induce the differentiation of PGCLCs from formative ESCs and EpiLCs at scale by overexpressing four TFs (*Prdm1*, *Prdm14*, *Tfap2c*, and *Nanog*). The resulting PGCLCs exhibit transcriptional and epigenetic profiles resembling those of late (E11.5) PGCs, and could be further differentiated into spermatogonia-like cells. Using this platform, we performed a CRISPR inhibition (CRISPRi) screen and identified several epigenetic factors that impacted the efficiency of PGCLC specification and proliferation, including *Ncor2*. Furthermore, we found that the exposure to Class I and Class IIa HDAC inhibitors attenuated PGCLC formation in vitro and germ cell formation in mice exposed during development. Our in vitro differentiation system holds promise for facilitating high-throughput functional genomic studies of germ cell development, characterization of potential infertility-causing genetic variants, and improving IVG.

# Results

## Establishing a cytokine-free platform for efficient production of PGCLCs from EpiLCs

To establish an efficient, cytokine-free system for generating PGCLCs, we began with a mouse ESC line containing enhanced GFP (eGFP) under the regulatory control of *Stella* (also called *Dppa3/Pgc7*), a PGC marker (Payer et al, 2006), and validated the ability of this line to form germline-transmitting chimeric mice after blastocyst microinjection (Fig. EV1A–D), demonstrating its pluripotency. The Stella-eGFP reporter enables real-time visualiza-tion of PGCLC development from ESCs in vitro. Next, based on the finding that *Nanog* promotes transition of ESCs to PGCLCs by binding the enhancer regions of *Prdm1* and *Prdm14* and inducing their expression (Murakami et al, 2016), we introduced a stably-integrated doxycycline (Dox)-inducible *Nanog*. To test the effec-tiveness of this transgene, we utilized a high-efficiency enhancer activity detection system (Kvon et al, 2020) to derive ESC lines containing targeted integrations of a lacZ reporter under the control of *Prdm1* or *Prdm14* enhancers into the safe harbor H11 locus. Dox treatment of embryoid bodies (EBs; ESCs>EpiLCs>EBs) derived from these lines triggered β-galactosidase production, indicating the effectiveness of NANOG-stimulated transcription (Fig. EV1E–H). Besides promoting expression of *Prdm1* and *Prdm14*, NANOG represses *Otx2*, a TF that promotes differentia-tion of pluripotent cells towards a neuroectodermal fate (Acampora

et al, 1995; Murakami et al, 2016). Mutant cells lacking OTX2 enter the germline from an EpiLC state with high efficiency, even in the absence of the essential PGCLC-promoting cytokine signals (Zhang et al, 2018a). In summary, NANOG plays a crucial role in PGCLC induction by promoting the expression of PGC core regulatory TF-encoding genes *Prdm1* and *Prdm14*, while simultaneously inhibiting the somatic fate inducer *Otx2* (Fig. EV1I).

Overexpression of *Prdm1*, *Prdm14*, and *Tfap2c* (hereafter called "3TF") can induce PGCLCs without cytokines, however, the efficiency of this differentiation system is relatively low with no increase in the fraction of cells becoming PGCLCs after day 4 post-induction (Nakaki et al, 2013; Magnúsdóttir et al, 2013). Because *Nanog* can induce PGCLCs independently of BMP signaling and performs better when combined with cytokines (Murakami et al, 2016), we hypothesized that simultaneous overexpression of NANOG and downstream targets (i.e., *Prdm1*, *Prdm14*, *Tfap2c*, and *Nanog*, hereafter called "4TF") in EpiLCs might render PGCLCs robust. We isolated several Dox-inducible clonal cell lines harboring combinations of these TFs in Stella-eGFP ESCs that also contained rtTAM2 (reverse tetracycline-controlled transactivator mutant; Fig. 1A,B). The resulting ESCs, cultured in 2i+LIF medium, were differentiated to EpiLCs by 2 days of Activin A and bFGF exposure. To induce PGCLC induction, the EpiLCs aggregates were treated with Dox instead of cytokines. We first tested whether ectopic expression of one or more TFs drove EpiLC-to-PGCLC induction in U-bottom 96-well plates. Overexpression of *Nanog* alone led to ~35% *Stella-eGFP*⁺ cells in day 6 EBs. This percentage increased when *Nanog* was combined with one of the core TF transgenes (i.e., *Nanog+Prdm1*, *Nanog+Prdm14*, and *Nanog+Tfap2c*), even though these genes are upregulated by NANOG during PGCLC induction (Murakami et al, 2016). Remarkably, simultaneous overexpression of all 4TFs caused ~80% of cells to become eGFP⁺ (Fig. 1A–E), representing a twofold increase in efficiency compared to the cytokine induction system using the same cell line (Hackett et al, 2018). Reverse transcription PCR (RT-PCR) using primers specific to the transgene TF constructs confirmed that all were induced by Dox (Fig. EV1J).

In vivo, PGCs undergo substantial epigenetic reprogramming, characterized by the loss of H3K9me2 and the acquisition of H3K27me3, as well as global erasure of DNA methylation (Seisenberger et al, 2012). To test the fidelity of epigenetic changes in PGCLCs produced in our 4TF system, we examined histone modifications, CpG methylation status of imprinted genes, and capacity for dedifferentiation into pluripotent EGCs. Differentiated cells with higher eGFP fluorescence displayed lower H3K9me2 and higher H3K27me3 labeling (Fig. 1F). The mRNA levels of epigenetic modifiers *Dnmt3a* and *Dnmt3b* were increased in EpiLCs and subsequently downregulated in EBs (Fig. 1G), which aligns with previous findings (Hayashi et al, 2011). Bisulfite sequencing of PGCLCs (eGFP⁺ cells) revealed reduced methylation in differentially methylated regions for both paternally (*H19*) and maternally (*Snrpn*) imprinted loci, while eGFP⁻ cells (presumably somatic-like) exhibited a more methylated state (Fig. 1H). In addition, PGCLCs could de-differentiated to EGCs in vitro when transferred to 2i+LIF medium (Appendix Fig. S1). Collectively, our data indicate that simultaneously overexpression of 4TFs effectively induces the differentiation of EpiLCs into PGCLCs.

## Transcriptomic properties of 4TF-induced PGCLCs

To examine the fidelity of our 4TF-induced PGCLCs, i.e., Stella-eGFP⁺ cells (Appendix Fig. S2), RNA sequencing (RNA-seq) was performed and the transcriptomes were compared to those of in vivo germ cells (encompassing PGCs at E9.5, E10.5, and E11.5 and male/female germ cells at E12.5 and E13.5) and cytokine (Ck)-induced PGCLCs (those at days 2, 4 and 6 after induction, referred to as Ck_D2/4/6PGCLC) (Yamashiro et al, 2016; Ohta et al, 2017; Sasaki et al, 2015). Principal component analysis (PCA) showed that 4TF PGCLCs at days 2, 4, and 6 (4TF_D2/4/6PGCLC) clustered tightly as a group separate from naive ESCs, EpiLCs and Ck_D2PGCLCs, representing a transient state towards the acquisition of a PGC-like state from EpiLC/epiblast states (Nakaki et al, 2013) (Fig. 2A). Notably, 4TF-induced PGCLCs grouped closely with PGCs at E10.5 and E11.5, while Ck_D4/6PGCLCs grouped closely with E9.5 and E10.5 PGCs (Fig. 2A). These data suggest that 4TF-induced PGCLCs are more akin to late PGCs than Ck_D2/4/6PGCLCs. Pearson correlation coefficient analysis revealed that 4TF_D2/4/6PGCLCs were highly correlated with Ck_D4/6PGCLCs but not Ck_D2PGCLCs (Fig. 2B), consistent with the PCA. We next applied two-sided rank–rank hypergeometric overlap (RRHO) analysis to compare the gene expression signatures between samples in a threshold-free manner (Cahill et al, 2018). We observed robust overlap in both up- and downregulated transcripts amongst differentiating cells in both Ck- and 4TF-induced groups relative to either ESCs or EpiLCs (Fig. 2C). In the Ck-induced group, ~60% of genes exhibited concordant differential expression (53% when compared to ESCs and 67% when compared to EpiLCs) between D2 and D6 PGCLCs. The concordance was higher for the D2 and D6 4TF-induced group (74% and 78%, respectively). In both induction systems, D4 and D6 PGCLC shared around 84% concordantly regulated genes (Fig. 2D). These data demonstrated that 4TF-induced PGCLCs at early stages exhibited greater consistency in their transcriptomic changes compared to those in Ck-induced PGCLCs. Given that only microarray (not RNA-seq) datasets were available for 3TF- and Nanog-induced PGCLCs, we performed a Spearman correlation coefficient analysis to compare their similarities with Ck-induced PGCLCs. Microarray datasets of 3TF- and Nanog-induced PGCLCs (referred to as 3TF_D2/4PGCLC and Nanog_D4PGCLC; datasets for 3TF_D6PGCLC and Nanog_D2/6PGCLC are not available) were compared to the corresponding microarray datasets of Ck-PGCLCs generated using the same platforms. The 3TF_D2/4PGCLCs showed high correlation with Ck_D4/6PGCLCs but not Ck_D2PGCLCs; Nanog_D4PGCLC showed a similar correlation to Ck_D4PGCLCs (Fig. 2B).

In vivo, PGC specification follows WNT-dependent induction, which activates the master mesodermal TF, T. This triggers the core TF network for PGCs (i.e., *Prdm1*, *Prdm14*, and *Tfap2c*). Whereas the Ck-induction system relies on WNT signaling, the 3TF and Nanog induction systems do not. To investigate WNT signaling activity during 4TF-induced PGCLC induction, we compared the transcriptomes of Ck- and TF-induced PGCLCs. Gene set variation analysis (GSVA) revealed strong enrichment of Wnt signaling pathways in Ck_D2PGCLCs (including downstream target genes *Wnt3*, *Wnt5a*, *Wnt5b*, *Axin2*, *Mixl1*, *Cdx2*, and *T*), as previously reported (Nakaki et al, 2013), but not in 3TF_D2PGCLCs or in our 4TF_D2PGCLCs (Fig. 2E,F). Overall, unlike Ck induction, the 4TF,

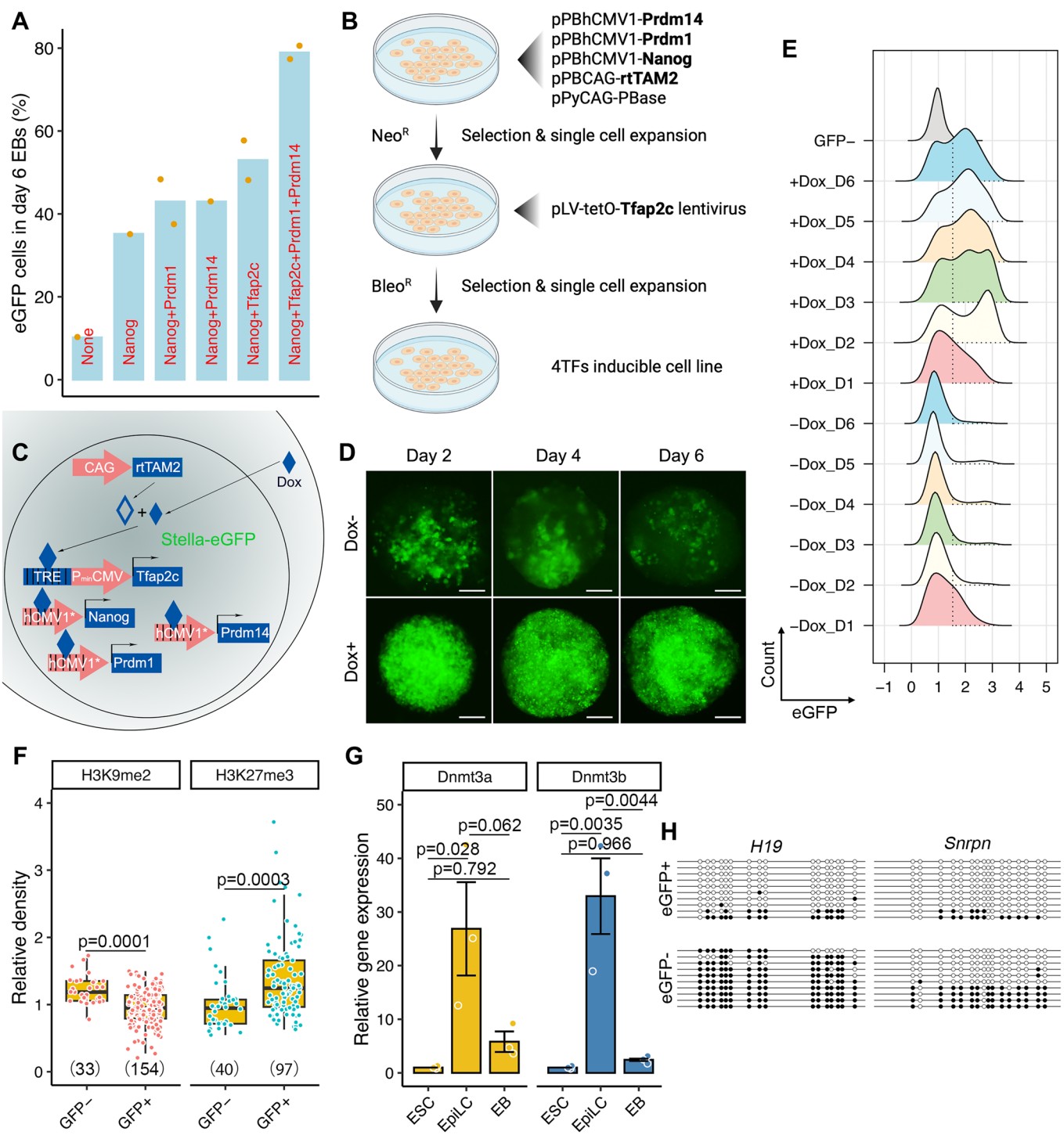

**Figure 1. Construction of the 4TF-induced PGCLC differentiation platform.**

(A) Percentages of day 6 PGCLCs (*Stella-eGFP⁺* cells) differentiated from Dox-inducible ESCs harboring distinct combinations of TFs. Each individual clonal line is represented by a dot. (B) The process for constructing 4TF-inducible ESCs. The 4TFs and rtTA are in bold. (C) Schema illustrating the 4TF-induced PGCLC differentiation from ESCs. (D) Representative EBs with eGFP signal in the presence or absence of Dox during the 6-day period. Scale bar in (D) represents 100 μm. (E) FACS patterns of *Stella-eGFP⁺* cells (indicated by dashed lines) during differentiation. (F) Relative fluorescence density of H3K9me2 and K3K27me3 in GFP⁻ and GFP⁺ cells from Dox-induced day 6 EBs. Black central line represents the median, and boxes and whiskers represent the 25th and 75th and 2.5th and 97.5th percentiles, respectively. (G) Relative mRNA expression of *Dnmt3a* and *Dnmt3b* in ESCs, EpiLCs, and day 6 EBs. (H) Bisulphite sequence analysis of methylated cytosine in the DMRs of the imprinted genes in *Stella-eGFP⁺* and *Stella-eGFP⁻* cells in day 6 EBs. White and black circles represent unmethylated and methylated cytosines, respectively. Data in (G) are represented as the mean ± SEM, n = 3 biological replicates. Data in (F) was analyzed using two-tailed unpaired *t* test and data in (G) was analyzed using one-way ANOVA with Tukey's post hoc test. Source data are available online for this figure.

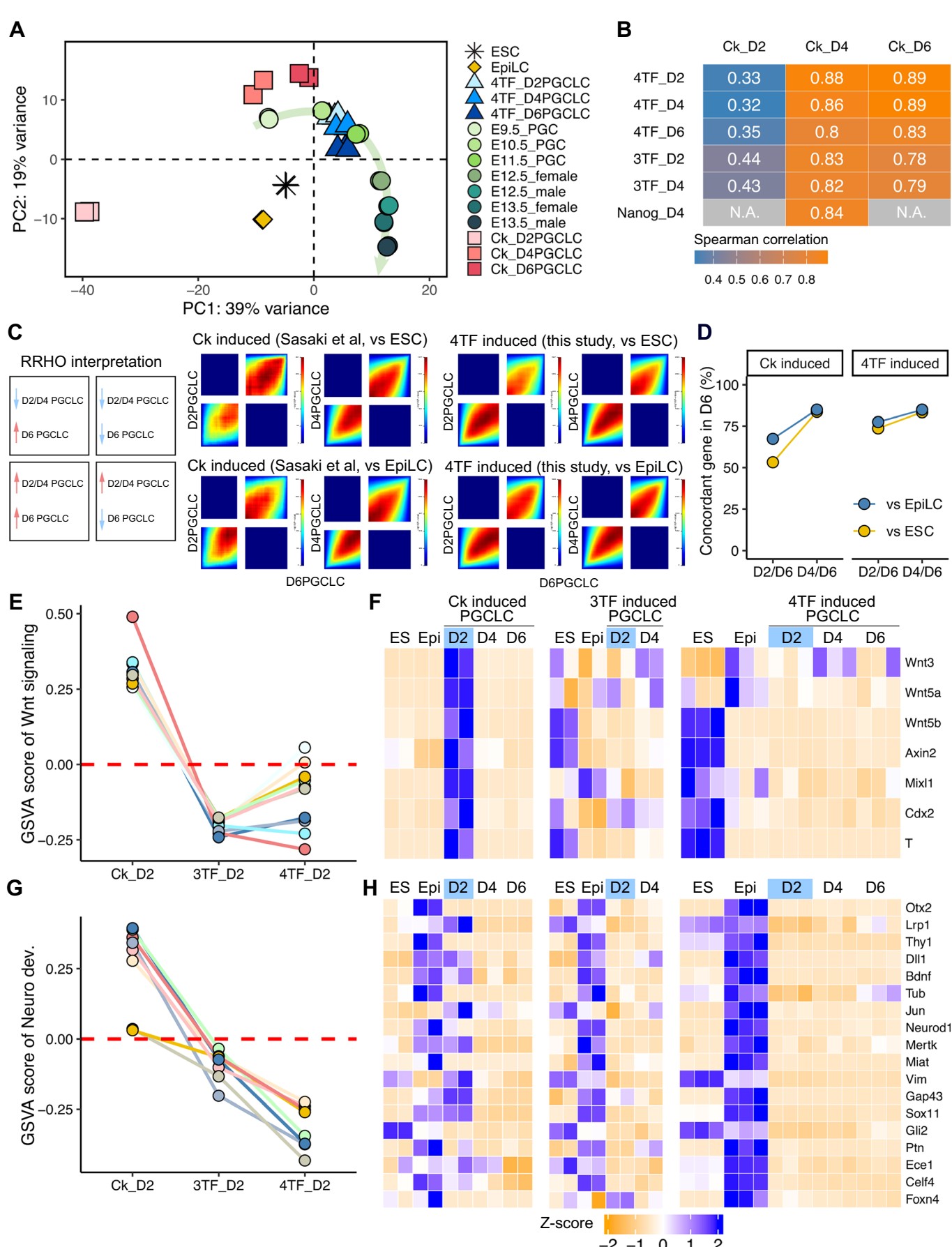

**Figure 2. 4TF-induced PGCLCs share similar transcriptomes with cytokine- and 3TF-induced PGCLCs.**

(A) PCA of indicated transcriptomes from 4TF- and cytokine (Ck)-induced PGCLCs, in vivo germ cells at indicated times of gestation, EpiLCs, and ESCs. "D" = days of induction. PGCLCs were isolated by flow sorted for diagnostic markers. (B) Spearman correlation coefficients between PGCLCs. 4TF_D2/4/6PGCLCs were compared with RNA-seq datasets of Ck_D2/4/6PGCLCs. 3TF_D2/4PGCLCs were compared with microarray datasets of Ck_D2/4/6PGCLCs, generated using the Affymetrix Mouse Genome 430 2.0 Array. Nanog_D4PGCLC was compared with the microarray dataset of Ck_D4PGCLC, generated using the Illumina MouseWG-6 v2.0 expression beadchip. N.A. not available. (C) Comparison of differential expression profiles between D6PGCLC (x axis) and D2/4PGCLC (y axis) with RRHO2 maps. The extent of overlap transcriptome was represented by each heatmap, colored based on −log10(P value) from a hypergeometric test. Each map shows the extent of overlapped upregulated transcripts, versus ESCs or EpiLCs, in the bottom left corner, whereas shared downregulated genes are displayed in the top right corners. (D) Percentage of concordant genes encompassing both up- and downregulated genes compared with D6PGCLCs. (E) GSVA score of Wnt signaling pathways (10 out of the 11 pathways in total were selected, with one specific to the planar cell polarity being excluded) in D2PGCLC between Ck-, 3TF-, and 4TF-induced conditions (Ck_D2, 3TF_D2, and 4TF_D2). ESC, EpiLC, and PGCLC in each group were used to calculate GSVA score and only the scores of D2PGCLC were displayed. (F) Heatmap of selected genes associated with WNT signaling and targets. (G) GSVA score of neurodevelopment-related pathways (see Appendix Fig. S3C) in D2PGCLC between Ck-, 3TF-, and 4TF-induced conditions (Ck_D2, 3TF_D2 and 4TF_D2). (H) Heatmap of selected genes associated with neurodevelopment. Source data are available online for this figure.

3TF, and Nanog induce PGCLCs from EpiLCs independently of the BMP–WNT-T pathway.

Unlike our 4TF system, the 3TF system exhibits low differentiation efficiency that did not increase after day 4 after induction (Nakaki et al, 2013; Magnúsdóttir et al, 2013). To explore the impact of *Nanog* overexpression in driving efficient differentiation into PGCLCs, we performed Gene set enrichment analysis (GSEA). This revealed that pathways predominantly related to neurodevelopment were suppressed in D2PGCLCs compared to EpiLCs (Appendix Fig. S3A,B), and *Otx2* was involved in the majority of these pathways (Appendix Fig. S3C). *Otx2* plays a crucial role in brain development and exhibits reciprocal antagonism with *Nanog*. Induction of transgenic *Nanog* in EpiLCs shifts the balance between these mutual antagonists, favoring *Nanog* (Vojtek et al, 2022). When *Otx2* levels decrease sufficiently, this enables NANOG to influence the regulatory elements controlling genes associated with PGC-specific TFs (Vojtek et al, 2022). GSVA revealed that neurodevelopment-related pathways and genes were highly repressed in 4TF_D2PGCLCs, while 3TF_D2PGCLCs showed a modest level of suppression. Notably, these pathways and genes were enriched in Ck_D2PGCLCs (Fig. 2G,H; Appendix Fig. S3D).

To better characterize the transcriptome landscapes during PGCLC induction, we applied k-Means analysis of top 500 differentially expressed genes (DEGs) and defined one cluster (C4) representing genes expressed in 4TF_D2/4/6PGCLCs (Fig. 3A). Gene Ontology (GO) analysis revealed that 11 of the top 20 pathways in this cluster were associated with reproductive processes (Fig. 3B). GSVA confirmed enrichment in germ cell development terms such as gamete generation, oogenesis, piRNA metabolic process, meiosis and reproduction were enriched in 4TF-induced but not Ck- and 3TF- induced PGCLCs compared to ESCs and EpiLCs (Figs. 3C; Appendix Fig. S4). GSEA revealed upregulation of genes related to gamete generation, meiotic cell cycle and reproduction in 4TF-induced PGCLCs. Processes related to piRNA processing and meiotic cell cycle also enriched in 3TF-induced D4PGCLCs. However, upregulation of these gene sets were barely observed in Ck- and Nanog-induced PGCLCs (Fig. 3D; Appendix Fig. S5). We next compared the DEGs among Ck-, 3TF-, and 4TF-induced PGCLCs. Across all three groups, the naive pluripotency gene *Klf4* was highly expressed in ESCs, while genes related to the formative state (*Otx2*, *Pou3f1*, *Fgf5*, *Dnmt3a*, and *Dnmt3b*) were highly expressed in EpiLCs (Fig. 3E). Early PGC-related genes (e.g., *Prdm1*, *Prdm14*, *Nanog*, *Dnd1*, *Tfap2c*, *Dppa3*, and *Itgb3*) were notably expressed in Ck_D4PGCLC,

3TF_D2PGCLC, and 4TF_D2PGCLC, indicating that both 3TF and 4TF-induction systems cause molecular differentiation more rapidly than cytokine induction. Some PGC genes, including *Rhox9*, *Rhox6*, *Rhox5* and *Gm9*, were highly expressed in Ck_D6PGCLC, 3TF_D4PGCLC and 4TF_D4PGCLC. Interestingly, late PGC, meiosis, and piRNA-related genes were consistently expressed in 4TF-induced PGCLCs but not in Ck-induced PGCLCs (Fig. 3E; Appendix Fig. S6). As expected, primed-state genes (*Sfrp2*, *Sox11*, and *Jakmip2*) were barely detectable in 4TF_D2/4/6PGCLC, with raw counts ranging only from 0 to 200. Network analysis identified a regulatory axis that facilitates the transition from early to late PGCLCs (Appendix Fig. S7; see "Discussion").

## Further differentiation of 4TF-induced PGCLCs in vitro

We next examined the potential of PGCLCs to differentiate into advanced stages in vitro. D6PGCLCs (*Stella-eGFP*+ cells) were aggregated with dissociated neonatal testicular cells from germ cell-deficient Kit^W/Wv mice in U-bottom 96-well plates. Then, the cell aggregates were transferred to a trans-well plate and cultured in medium containing GNDF and bFGF that are essential for promoting spermatogonial growth (Kubota et al, 2004) (Fig. 4A). After a 2-week induction period, we conducted immunofluorescence (IF) staining to assess whether the aggregates express late PGC or spermatogonia proteins. We detected cell clusters that expressed NANOS3, DAZL, and DDX4 within the aggregates (Fig. 4B). Over 70% of DDX4-positive cells also expressed GCNA (Germ cell nuclear antigen) (Fig. 4C), a robust marker of germ cells from E11 onwards (Enders and May, 1994; Carmell et al, 2016). Interestingly, an average of 8.4% of DDX4-positive cells also contained the spermatogonial marker PLZF (formally ZBTB16; Zinc finger and BTB domain-containing protein 16) (Buaas et al, 2004) after 2 weeks of co-culture, whereas this was not observed after 1 week of culture. As expected, DDX4-positive cells did not express the Sertoli cell marker SOX9 (Fig. 4C). To test whether the germ cells could develop to more advanced stages, the aggregates were cultured using a gas-liquid interphase method for an additional week in the medium containing KSR (knockout serum replacement) at 34 °C (Fig. 4A) (Sato et al, 2011). Interestingly, cells expressing both GCNA and STRA8 were identified in the aggregates (Fig. 4D). These observations suggest that 4TF-induced PGCLCs have some capacity for further development in vitro, but optimization of conditions and functional assays will be required to determine their full potential.

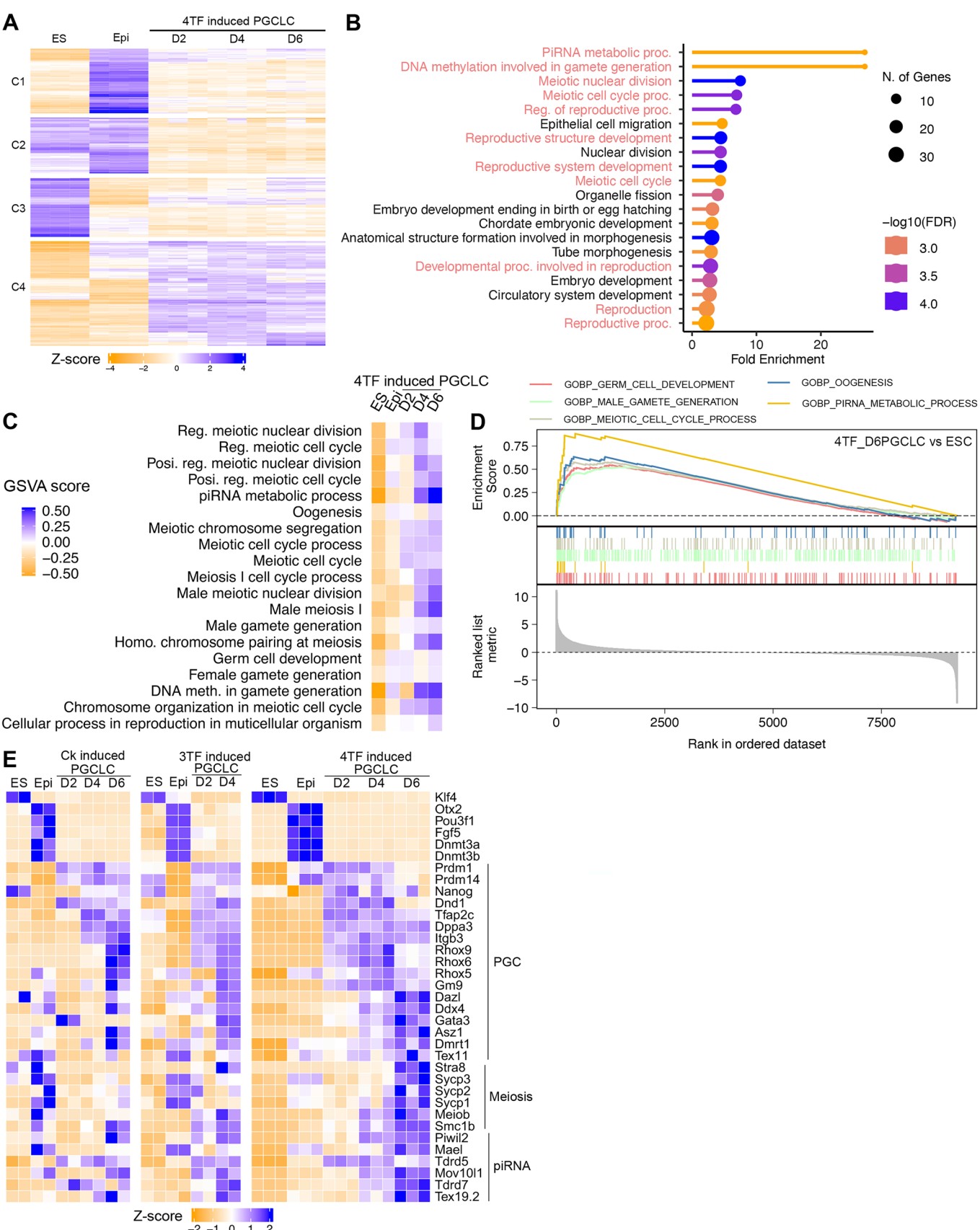

Figure 3. Gene expression profiling of 4TF-induced PGCLCs.

(A) Heatmap of k-Means clustering of variably expressed genes in ESCs, EpiLCs, and 4TF-induced D2/4/6PGCLCs ($n = 500$, $k = 4$). Genes were grouped into four clusters ("C1-4") on the basis of expression similarity. (B) Biological process GO term analysis of Cluster 4 (C4). Terms of germ cell development (highlighted by red) was highly enriched for terms in C4. (C) Heatmap showing the average GSVA enrichment score of selected germ cell development pathways. (D) Representative GSEA enrichment plots depicting significant enrichment of germ cell signatures in D6PGCLCs compared with ESCs. (E) Heatmap of representative gene expression of pluripotency and germ cells among ESCs, EpiLCs, and PGCLCs.

## Scale-up differentiation and long-term culture of PGCLCs

To validate the Stella-eGFP transgene as an accurate marker of PGCLCs, we analyzed differentiating cultures with two additional PGC surface markers, ITGB3 (also known as CD61) and SSEA1 (Hayashi et al, 2011). Doubly positive cells were shown to differentiate into functional gametes (Hayashi and Saitou, 2013). As expected, day 6 EBs were significantly enriched for ITGB3+/SSEA1+ and ITGB3+/Stella-eGFP+ cells compared to ESCs and EpiLCs (Fig. 5A). Next, we sought to determine the scalability of 4TF-induction system. We seeded $2 \times 10^3$ cells/ well to U-bottom 96-well plates, $8 \times 10^4$ cells/well to untreated 12-well plates (suitable for suspension culture) and $1 \times 10^6$ cells to a single 100 mm bacteriological Petri dish, then assessed differentiation in day 6 EBs. The percentages of both ITGB3+/SSEA1+ and ITGB3+/Stella-eGFP+ populations were highest in 12-well plates and lowest in 96-well plates (Fig. 5A,B). Suspecting that EB size influences differentiation efficiency, we measured the diameters of day 6 EBs. In the U-bottom 96-well plates, EB sizes were homogeneous and by far the largest (~2.5× on average) amongst the culture formats (Fig. 5C). The results indicate an inverse relationship between efficiency of PGCLC induction and EB size, which is consistent with observation in the human PGCLC induction system (Jo et al, 2022). The scalability of the Nanog-only induction system was also observed by seeding varying numbers of EpiLCs into different plates/dish. However, the percentages of both ITGB3+/SSEA1+ and ITGB3+/Stella-eGFP+ populations were 30% and 60% lower, respectively, than those in the 4TF-induction system (Appendix Fig. S8A; 12-well plate cultures). EpiLCs represent a transient, germline-competent formative state, with day 2 EpiLCs demonstrating the highest efficiency in response to cytokine induction compared to day 1 or 3 EpiLCs (Hayashi et al, 2011). We found that 4TFs can also induce PGC fate in day 1 EpiLCs, albeit with lower efficiency than in day 2 EpiLCs (Fig. 5D,E). Nanog overexpression alone was insufficient to differentiate day 1 EpiLCs into PGCLCs (Appendix Fig. S8B,C).

Next, we investigated the potential for maintaining PGCLCs over long periods. Day 6 EBs from both 4TF- and Nanog-induction systems were dispersed, and $8 \times 10^4$ cells/well were seeded into a 12-well plate. The cells were passaged every 4 days in suspension (Fig. 5F). We successfully maintained the EBs for 10 continuous passages with Dox-induced transgene expression. However, cell numbers declined without Dox in both systems, indicating that TFs, particularly Nanog, are essential for sustaining EB cell proliferation (Fig. 5G). While the doubling times between 4TF- and Nanog-induced cells were comparable (Fig. 5H), only the 4TF system consistently maintained the PGC identity (>70% of cells), whereas nearly all cells in the Nanog-only system lost PGC marker expression (Figs. 5I; Appendix Fig. S8D,E). Lastly, we investigated the impact of culture plate type on PGCLC induction by seeding

$8 \times 10^4$ cells/well onto treated 12-well plates designed for adherent culture. After differentiation, we observed the presence of attached cells, along with both floating and attached EBs (Appendix Fig. S9A). While PGCLCs were robustly induced, the differentiation efficiency was lower compared to those cultured in untreated plates (Appendix Fig. S9B,C), indicating that suspension culture is more effective in promoting PGCLC induction compared to adherent culture.

## High-throughput CRISPRi perturbation screen identifies epigenetic regulators impacting PGCLC formation

A major motivation for developing an inexpensive, simple, and scalable germ cell culture system was to conduct genome-scale functional genomic screens for genes involved in various aspects of germ cell biology and development. Epigenetic reprogramming is an especially important aspect of normal PGC development. To identify epigenetic regulators impacting PGC development, we designed a CRISPRi screening strategy to target all the genes with known or putative functions in epigenetic regulation (Fig. 6A). We opted to use a CRISPRi rather than a standard knockout-oriented CRISPR-Cas9 strategy, reasoning that complete ablation of some epigenetic genes would be generically disruptive to cell proliferation, and not representative of genetic or environmental perturbations. We built a custom lentiviral library encoding 7360 sgRNAs targeting 701 "epigenetic" genes, plus 350 non-targeting sgRNAs as controls (Datasets EV1 and EV2), and introduced these into a 4TF-inducible ESC line containing an integrated dCas9-KRAB (Krüppel-associated box) transgene (Appendix Fig. S10A–D). The library-infected ESCs contained 89% of all the sgRNAs in the library, with at least 5 sgRNAs present for 694 genes (Appendix Fig. S10E and Dataset EV3). PGCLC induction was initiated in 100 mm bacteriological Petri dishes (Fig. 6A), and both Stella-eGFP+ and Stella-eGFP- cells were isolated from dispersed day 6 EBs (Appendix Fig. S10F). In each resulting population and ESCs, sgRNAs were PCR amplified and then sequenced. Quality control measures indicated a high degree of uniformity and coverage of the sgRNA library in each cell population (Appendix Fig. S11). The MAGeCKFlute algorithm was utilized to calculate a beta score for every potential gene by comparing each population to the ESCs (Wang et al, 2019). Genes with a negative beta score suggest negative selection of the epigenetic factor within the cell population, whereas a positive beta score signifies positive selection. Using one standard deviation from the mean as a threshold, we identified 53 and 61 under- and over-represented genes, respectively, from the 701 queried genes (Fig. EV2A,B; Dataset EV4).

Transduction of sgRNAs targeting *Ncor2*, which encodes a nuclear receptor co-repressor that facilitates histone deacetylation (Nagy et al, 1997; Huang et al, 2000), had the most negative impact on 4TF-induced PGCLC in our screen (Fig. EV2; Dataset EV4).

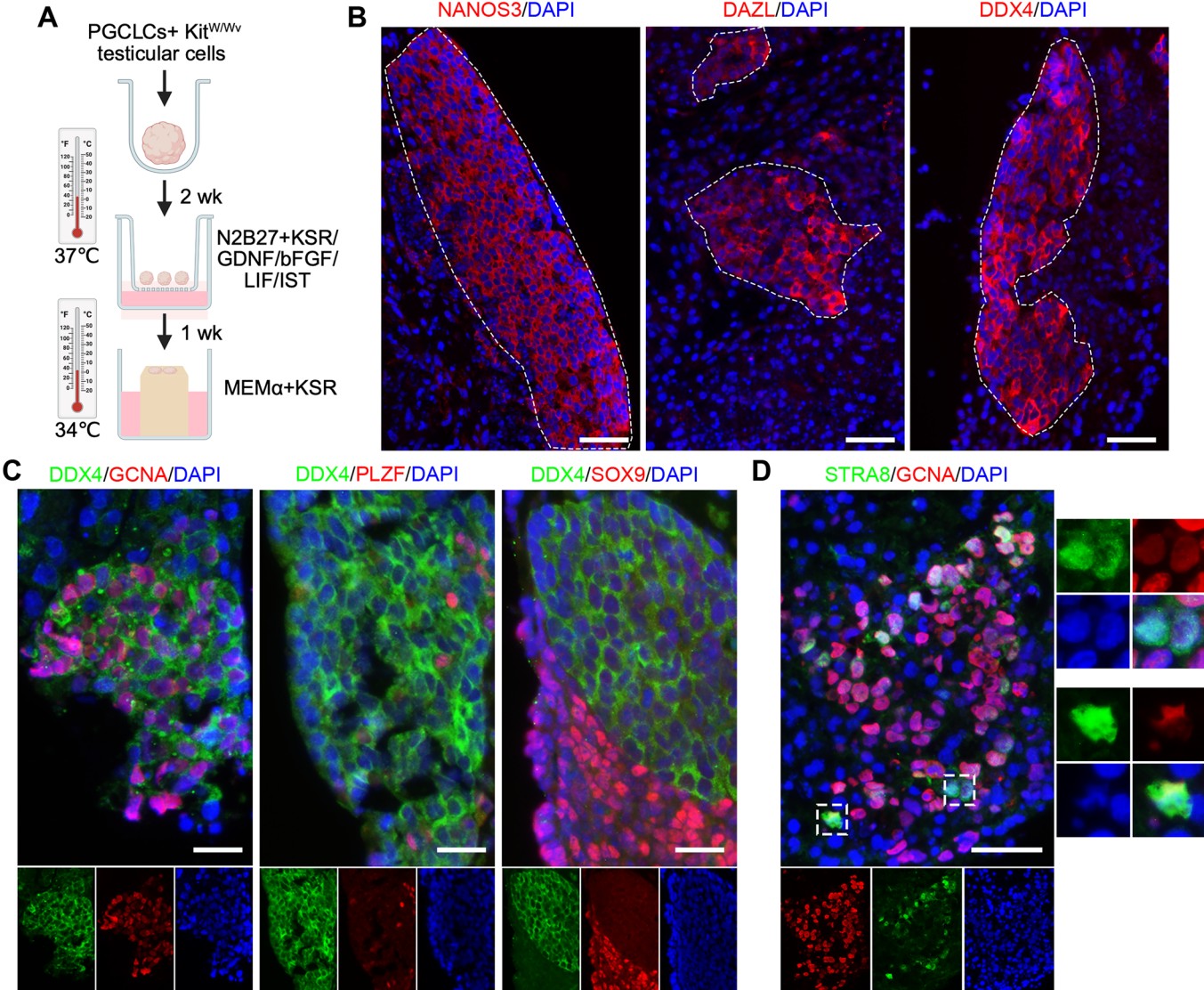

**Figure 4. Development of cells past PGCLC stage.**

(A) Further development of 4TF_D6PGCLCs by aggregation with testicular somatic cells isolated from Kit^w/wv mice. The cultured aggregates were subjected to IF. (B) Clusters of germ cells labeling by NANOS3, DAZL and DDX4 were identified from aggregates cultured after 2 weeks. Dashed lines denote regions with germ cell clusters. (C) Dual IF staining of DDX4 with GCNA, PLZF, and SOX9 in the aggregates after 2-week culture. (D) Dual IF staining of GCNA and STRA8 in the aggregates cultured for an additional week. The boxed region is magnification of representative dual positive cells. Scale bars in (B–D) represent 50 μm. Source data are available online for this figure.

Treatment with *Ncor2*-targeting siRNA caused significant reductions in *ITGB3⁺/Stella-eGFP⁺* cells (Fig. EV3A–D), consistent with a previous observation from a Ck-induction system (Mochizuki et al, 2018). NCOR2 contains three repression domains (RD1, RD2, and RD3) capable of interacting with various classes of HDACs (Guenther et al, 2000) (Fig. EV3E). Because one motivation of this project was to identify environmental agents (including drugs and dietary supplements) that might cause deleterious epigenetic germline perturbations, we tested the effects of two HDAC inhibitors on PGCLC development: sodium butyrate (SB) and valproic acid (VPA). Both are short-chain fatty acids that inhibit HDACs by binding to the zinc-containing domains of these enzymes (Fig. EV3E) (Kim and Bae, 2011). Treatment with VPA or

SB resulted in a reduction in PGCLC induction and increased H3K27ac levels (Fig. EV3F–I).

## HDAC inhibition increases the expression of genes involved in somatic lineages

To investigate the mechanism underlying the reduced PGCLC induction efficiency following HDACi treatment, we conducted RNA-seq to analyze the transcriptomes of *Stella-GFP⁺* and *Stella-GFP⁻* cells on day 6 of differentiation from control and HDACi-treated groups (Fig. EV4A). PCA revealed that day 6 VPA-treated Stella-eGFP⁻ cells (VPA_eGFP⁻) and SB_eGFP⁻ cells clustered separately from the eGFP⁺ cohorts (VPA_eGFP⁺, SB_eGFP⁺ and

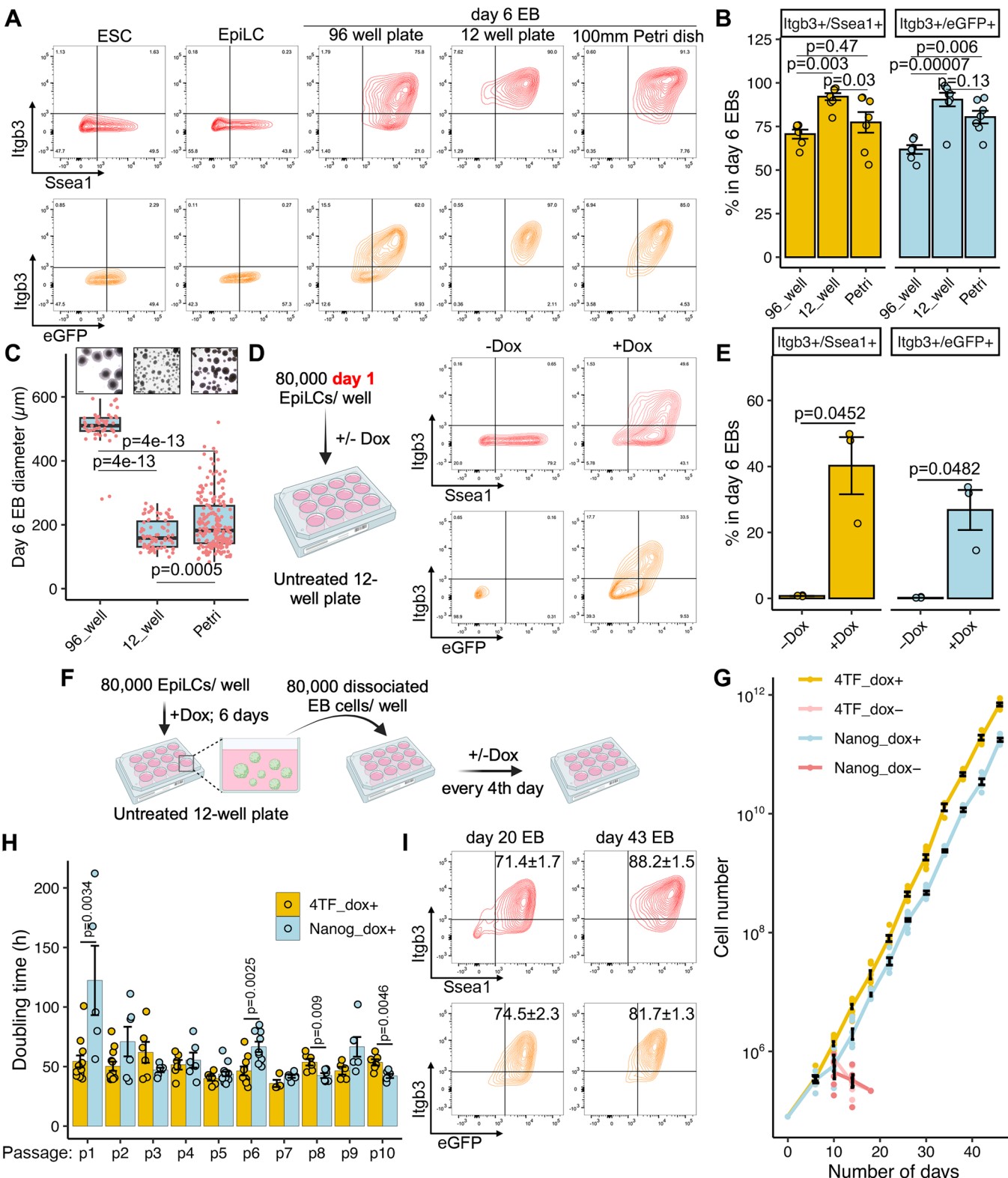

◄ **Figure 5. Scale-up and long-term differentiation of 4TF-induced PGCLCs.**

(A) Representative FACS pattern of Itgb3$^+$/Ssea1$^+$ and Itgb3$^+$/eGFP$^+$ cells in ESCs, EpiLCs, and day 6 EBs from U-bottom 96-well plates, 12-well non-treated plates and 100 mm bacteriological Petri dishes. eGFP, Stella-eGFP. (B) Quantification of PGCLC populations in day 6 EBs at different plates/dish. $N = 12$, 16, and 14 for the 96-well, 12-well and 100 mM Petri dish groups, respectively. (C) Boxplots showing the distribution of the diameter of day 6 EBs in different plates/dish. $n = 3$ biological replicates. Black central line represents the median, and boxes and whiskers represent the 25th and 75th and 2.5th and 97.5th percentiles, respectively. (D) Induction of 4TF-induced PGCLCs from day 1 EpiLCs. (E) Quantification of 4TF-induced PGCLC populations in day 6 EBs in the presence or absence of Dox. Day 1 EpiLCs were used for EB formation in non-treated 12-well plates. $n = 3$ biological replicates. (F) Schema depicting the process of EB passage. The dissociated EB cells were passaged every 4 days into each well of non-treated 12-well plates. (G) Growth curves of long-term cultured EBs induced by 4TFs, Nanog, or not. $n \geq 4$ biological replicates. (H) Doubling time of passaged EBs in 4TF- and Nanog-induced PGCLC differentiation systems. $n \geq 5$ biological replicates. (I) FACS pattern of Itgb3$^+$/Ssea1$^+$ and Itgb3$^+$/eGFP$^+$ cells induced by 4TFs in day 20 and 43 EBs cultured in 12-well non-treated plates. Data in (B, E, G, H) are represented as the mean ± SEM. Data in (B, C) were analyzed using one-way ANOVA with Tukey's post hoc test. Data in (E) were analyzed using a two-tailed paired $t$ test. Data in (H) were analyzed using a two-tailed unpaired $t$ test. Scale bar in (C) represents 200 μm. Source data are available online for this figure.

Ctrl_eGFP$^+$). However, among eGFP$^+$ cells, those treated with VPA clustered apart from the SB and control groups (Fig. EV4B). k-Means analysis of top 2,000 DEGs and defined three clusters (C1, C2 and C3) (Fig. EV4C). The genes in clusters C1 and C2, which were particularly elevated in eGFP$^-$ cells, were enriched with terms related to immune responses, cell migration and neural development, whereas cluster C3 containing predominantly eGFP$^+$ cells were enriched for genes involved in processes related to reproduction such as gamete generation and meiosis (Fig. EV4C). GSVA revealed that eGFP$^-$ cells from both HDACi-treated groups (VPA_eGFP$^-$ and SB_eGFP$^-$) displayed lower enrichment of Gene Ontology Biological Processes (GOBP) terms related to reproduction and germline development (Fig. EV4D). By contrast, eGFP$^+$ cells from control and treatment groups exhibited significant enrichment in all reproductive pathways, though the enrichment was greater in the untreated control group (Fig. EV4D). Despite HDACi treatment, genes related to early PGC development were normally expressed in eGFP$^+$ cells. However, they were downregulated in eGFP$^-$ cells. For genes tied to meiosis and piRNA metabolic processing, their expression decreased under HDACi treatment, even in the eGFP$^+$ cells (Fig. EV4E). These phenomena raise the possibility that failure to properly deacetylate (downregulate) non-germ cell regulatory elements may promote alternative lineage fates, a hypothesis tested by experiments described below.

To further explore the identity and fate of cells exposed to HDACi, we focused on upregulated DEGs in VPA_eGFP$^+$ and SB_eGFP$^+$ cells compared to Ctrl_eGFP$^+$ cells (Fig. EV5A,B). Gene set enrichment analysis (FDR < 0.05) of 211 overlapping genes revealed clusters of upregulated GOBP driver terms in VPA_eGFP$^+$ and SB_eGFP$^+$ cells such as placental development, muscle cell differentiation, and innate immunity responses (Fig. EV5C), the latter of which has been observed in macrophages and dendritic cells (Roger et al, 2011). The emergence of pathways related to placental and reproductive structure development, and muscle and cytoskeleton development suggests possible incomplete suppression of factors involved in extraembryonic and embryonic mesoderm lineage differentiation.

Lacking precise knowledge of the mechanism governing mesoderm lineage activation with HDACi treatment, we hypothesized that eGFP$^+$ cells in treatment groups would have a transcriptome intermediate between germline and somatic fates. To test this, we applied GSEA of upregulated DEGs in HDACi-treated eGFP$^-$ cells compared to their eGFP$^+$ counterparts (Fig. EV5D,E). The overlap genes enriched in GOBP driver terms

such as anatomical structure development, regulation of innate immune response, and homophilic cell adhesion. The first category was enriched for GO terms associated with the development of the placenta and all three germ layers, including embryonic, cardiac, paraxial (muscle, skeleton, and cartilage), intermediate (kidney, nephron, and urogenital), and lateral (mesenchyme, blood vessel, and vasculature) mesoderm. Pathways associated with endoderm (lung, respiratory system) and ectoderm (skin and brain) development were also enriched (Fig. EV5F). The results suggest that gene activation driven by histone hyperacetylation promotes overall somatic-lineage differentiation from epiblast cells.

## HDAC inhibitors impact PGC development

To assess the impact of histone deacetylation on in vivo PGC development, HDACi was administered to pregnant mice, and their offspring were examined for potential reproductive defects. From E5.5 to E11.5, pregnant females received daily injections of either 600 mg/kg or 800 mg/kg of SB per body weight or 300 mg/kg of VPA per body weight (Fig. 6B). All treated dams were viable and produced a comparable number of pups to the water-injected control group. Quantification of follicle numbers in 1-month-old ovaries revealed no significant differences compared to the controls (Appendix Fig. S12). In 2-month-old males, testis weights were unaffected by HDACi treatment during pregnancy (Fig. 6C). However, these animals exhibited a 30% reduction in sperm numbers in the SB groups and an 18% decrease in the VPA group (Fig. 6D). Histological analysis identified atrophic seminiferous tubules (1–3 per section of a whole testis), despite normal morphologies in the cauda epididymides (Fig. 6E). TUNEL (terminal deoxynucleotidyl transferase dUTP nick end labeling) staining revealed no apoptotic cells within atrophic tubules (Appendix Fig. S13). These findings suggest that fetal exposure to SB and VPA leads to decreased sperm counts and increased atrophic seminiferous tubules in adulthood, possibly a consequence of perturbations to PGCs or SSCs.

## Direct differentiation of PGCLCs from formative 4TF-inducible ESCs

To determine if 4TFs can efficiently induce PGCLCs from formative ESCs, a pluripotent stem cell that directly responds to PGC specification, the 4TF-inducible ESCs were transformed to a formative state through exposure to bFGF, Activin A and GSK3β inhibitor (CHIR99021) (Fig. 7A) as described previously (Yu et al,

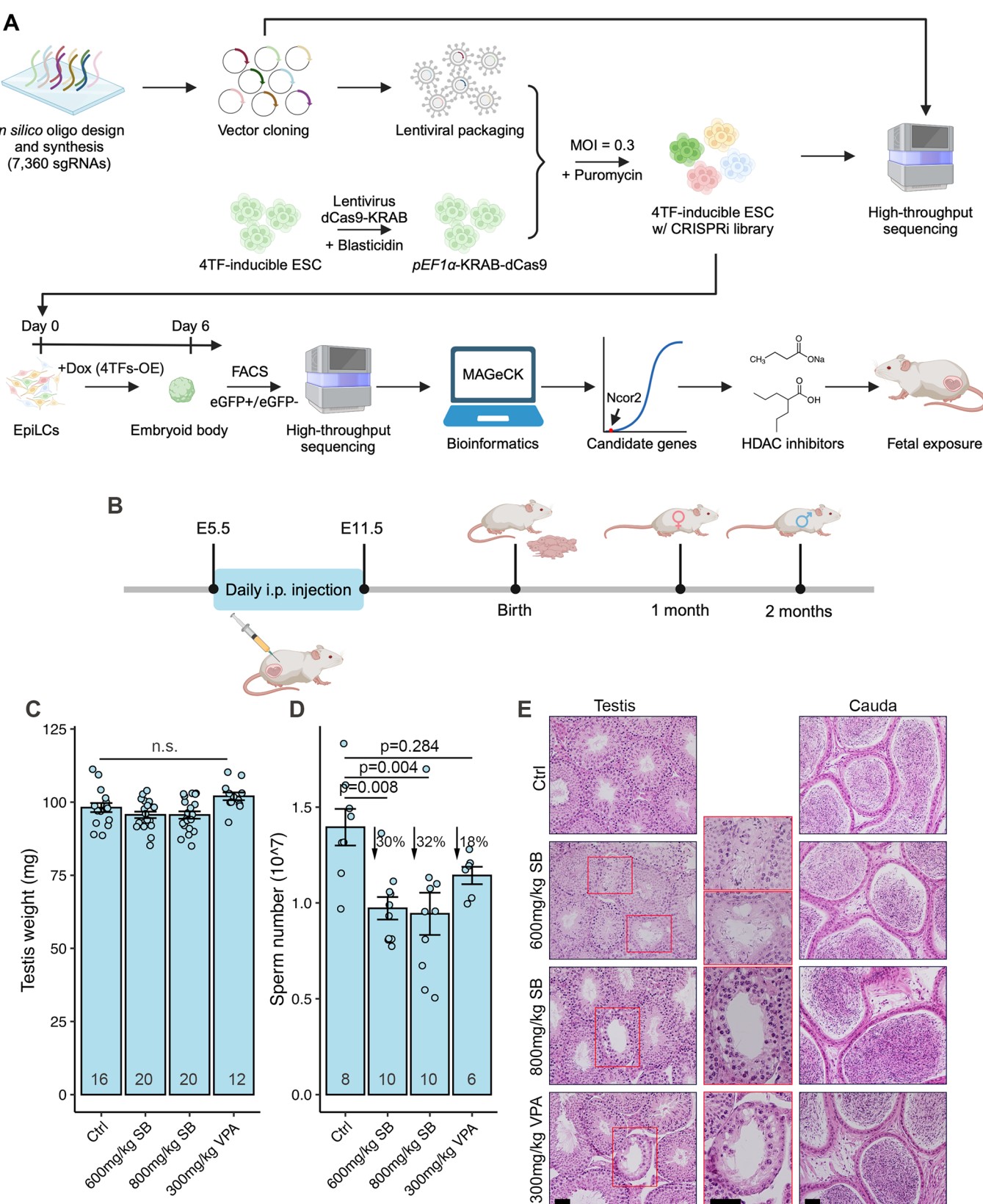

Figure 6.   Application of 4TF-inducible system for high-throughput screening epigenetic regulators.

(A) Schema of CRISPRi library generation and validation is shown on the top. The uniformity of sgRNA library representation was tested by sequencing of batch-cloned plasmids and lentivirus-infected ESCs (colored cell clusters). On the bottom is a schema of the PGCLC differentiation protocol and CRISPRi screening process. Day 6 EBs were sorted into Stella-eGFP⁺ and Stella-eGFP⁻ cells to generate count representatives for downstream analysis. Candidate gene *Ncor2* and HDAC inhibitors SB and VPA were selected for validation. OE, overexpression. (B) Experimental schema showing SB and VPA treatment on pregnant mice and analysis their pups. i.p., intraperitoneal. $N = 5$ for each group. (C) Testis weight of 2-month-old pups. n.s., no significant difference. (D) Sperm counts. (E) Histological analyses of 2-month-old testes and cauda epididymis. The boxed region are magnifications of atrophic seminiferous tubules. Data in (C, D) are represented as the mean ± SEM and were analyzed using one-way ANOVA with Tukey's post hoc test. Scale bars in (E) represent 50 μm. Source data are available online for this figure.

2021). Naive ESCs typically grow as tightly packed colonies with a dome shape. Upon EpiLC induction, the cells quickly underwent a morphological conversion that includes flattening, diminished cell-cell interactions, and the formation of cellular protrusions. However, formative ESCs displayed distinctive morphologies (Fig. 7B). Immunolabeling of ZO1 (zonula occludens-1, also known as tight junction protein 1) revealed that formative ESCs and EpiLCs were more similar to one another than naive ESCs (Fig. 7B). The regulation of *Oct4* expression in naive and primed pluripotent cells is differentially controlled by distal (DE) and proximal enhancers, respectively (Choi et al, 2016). The DE also exhibits activity in the formative state, albeit at a lower level compared to that in the naive state (Yu et al, 2021). To monitor the exit from the naive state during naive-to-formative conversion, we compared the GFP signal intensity of Oct4-DE-GFP ("OG2") iPSCs during this transition. Consistent with previous observations (Yu et al, 2021), the majority of formative iPSCs and EpiLCs were GFP⁺, but the average signal intensity was lower than that in naive iPSCs (Fig. 7B). Collectively, these results confirm the establishment of formative ESCs/iPSCs from naive cells.

We next tested if formative 4TF-inducible ESCs respond directly to PGC specification. After removing the feeder layer, PGCLC induction was conducted by seeding $8 \times 10^4$ ESCs/well to untreated 12-well plates (Fig. 7A). FACS analysis of day 6 EBs revealed distinct ITGB3⁺/SSEA1⁺ and ITGB3⁺/Stella-eGFP⁺ populations with induction by Dox; such cells were not observed in undifferentiated formative ESCs and without induction by Dox (Fig. 7C,D). However, the efficiency was lower compared to that observed when EpiLC were induced to PGCLCs (Fig. 5A,B). Treatment with an FGF receptor (FGFR) inhibitor PD173074 was reported to significantly increase Ck-induced PGCLC formation from formative ESC via unknown mechanisms (Yu et al, 2021). However, this enhancement was not observed in 4TF-induced PGCLCs (Appendix Fig. S14). Interestingly, overexpression of Nanog alone is insufficient to direct formative ESC toward a PGC fate (Appendix Fig. S15). We next aggregated the D6PGCLC (*Stella-eGFP⁺* cells) with somatic cells from Kit^{W/Wv} testes to assess further differentiation potential. Similar to EpiLC-derived PGCLCs (Fig. 4A), the formative ESC-derived PGCLCs demonstrated the capacity for further development (Fig. 7E). Collectively, these findings demonstrated that overexpressing 4TFs can directly and effectively induce differentiation of formative ESCs into PGCLCs.

To elucidate the differences in efficiency in generating PGCLCs from formative 4TF ESCs vs. day 2 EpiLCs, and to explore why Nanog alone is insufficient to drive differentiation of formative ESCs to PGCLCs, we compared the transcriptomes of our derived EpiLCs to those in a published dataset of formative ESCs cultured and differentiated under identical conditions (Lyu et al, 2024).

RRHO analysis indicated substantial similarities in gene expression changes during differentiation and a significant correlation between 48 h EpiLC and naive ESC transcriptomes, indicating a shared cell state (Fig. 7F). Interestingly, while formative ESCs overlapped with EpiLCs at various stages in co-upregulated genes, they did not show overlap in co-downregulated genes (Fig. 7F). This suggests that formative ESCs represent distinct sub-stages, leading to differences in PGCLC induction efficiencies.

## Discussion

PGC development entails important regulatory and epigenetic events that lay the groundwork for subsequent sex-specific gametogenesis programs. Although in vitro PGCLC induction systems have facilitated molecular studies of the specification and developmental events of the germ cell lineage (Aramaki et al, 2013; Nakaki et al, 2013; Murakami et al, 2016; Zhang et al, 2018a; Tischler et al, 2019; Hackett et al, 2018; Naitou et al, 2022; Yin et al, 2020), current methodologies have shortcomings in terms of efficiency, scalability and long-term culture. Our 4TF-induction system addresses these limitations, having several advantages compared to Ck-, 3TF-, and Nanog-only inducible systems (Appendix Fig. S16). We demonstrated that simultaneous overexpression of *Nanog*, *Prdm1*, *Prdm14*, and *Tfap2c* in mouse EpiLCs or formative ESCs can render PGCLCs in a highly efficient and cost-effective manner compared to traditional methods (~ 500× less expensive; Appendix Fig. S16). The induced PGCLCs exhibit late PGC hallmarks, and they can develop into spermatogonium-like cells in vitro. Overexpression of *Nanog* had the important effect of enhancing the PGC regulatory network, suppressing the formation of non-germline lineages, and maintaining PGC fate, significantly improving PGCLC production.

Specification of mouse PGCs requires coordinated actions of *Prdm1*, *Prdm14*, and *Tfap2c*. Here, we found that forced overexpression of *Nanog* combined with one of these core TFs can drive PGCLC specification more efficiently that *Nanog* alone, and 4TFs yielded the highest efficiency, indicating additive or synergistic effects. Given that NANOG can bind and activate enhancers of *Prdm1* and *Prdm14* in EpiLCs in vitro (Murakami et al, 2016), as we confirmed here, we conclude that *Nanog* overexpression strengthens the PGC core regulatory network in the 4TF system. NANOG also plays a role in suppressing somatic lineage induction development. For example, OTX2, a crucial TF for anterior neuroectoderm specification, shows reciprocal antagonism with NANOG and thus restricts mouse germline entry (Zhang et al, 2018a). 4TF-induced PGCLCs exhibited *Otx2* downregulation, *Nanog* upregulation, and significant downregulation of

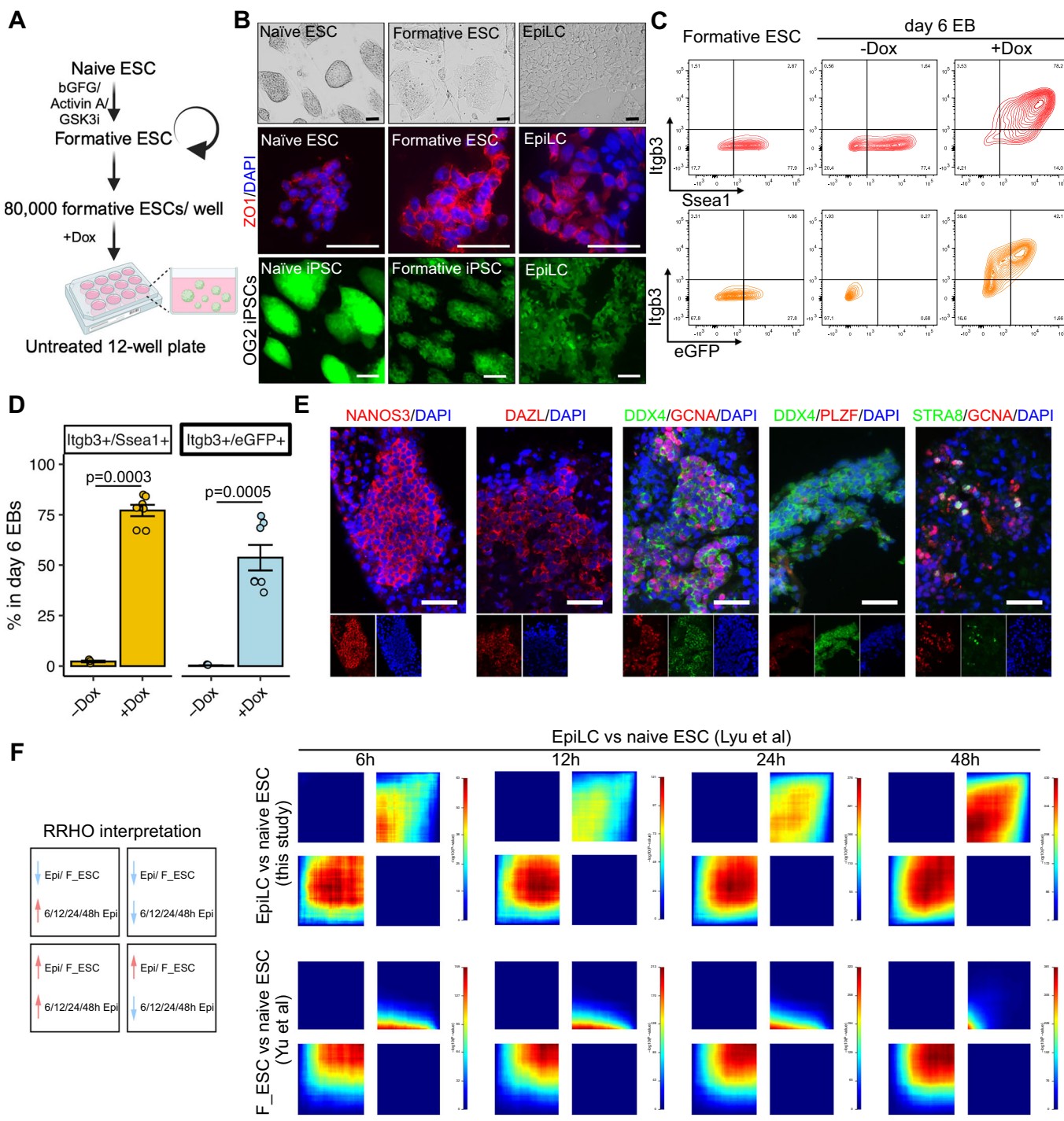

**Figure 7. Direct differentiation of PGCLCs from formative 4TF-inducible ESCs.**

(A) Schema for inducing PGCLCs from formative ESCs. (B) Characterization of formative ESC derived from naïve ESCs. Top panel: representative morphologies of naïve ESCs, formative ESCs and EpiLCs. Middle panel: IF staining of tight junction protein ZO1 on naïve ESCs, formative ESCs and EpiLCs. Bottom panel: GFP expression in naïve and formative OG2 (Oct4-DE-GFP) iPSCs, and EpiLCs differentiated from naïve OG2 iPSCs. (C) Representative FACS pattern of Itgb3⁺/Ssea1⁺ and Itgb3⁺/Stella-eGFP⁺ cells in formative ESCs and day 6 EBs with or without Dox treatment in 12-well non-treated plates. (D) Quantification of PGCLC subpopulations after 4TF induction. $n \geq 3$ biological replicates. (E) IF staining of germ cell markers in cultured D6PGCLCs (*Stella-eGFP*⁺) differentiated from formative ESCs aggregated with testicular cells from *Kit*$^{w/wv}$ mice. (F) RRHO analysis of transcriptomes from 4TF formative ESCs and EpiLCs compared to those from a published study (Lyu et al, 2024). Sampled time points during differentiation are indicated (6 h, 12 h, 24 h, and 48 h). Data in (D) are represented as the mean ± SEM and analyzed using a two-tailed paired *t* test. Scale bars in top and bottom panels of (B) represent 100 μm, and in middle panel of (B) and (E) represent 50 μm. Source data are available online for this figure.

neurodevelopment pathways in D2PGCLCs *vs* EpiLCs. However, these pathways were activated in Ck-induced PGCLCs or not fully repressed in 3TF-induced PGCLCs. Using *Nanog-eGFP* transgene iPSCs, Hayashi et al, demonstrated that *Nanog* was upregulated as early as day 2 in Ck-induced PGCLCs (Hayashi et al, 2011). The distinct kinetics of NANOG expression in the 4TF and Ck-induction systems correlated with the repression and activation of neurodevelopment pathways, respectively. Lastly, NANOG is essential for maintaining PGC fate. Depletion of *Nanog* in PGCLCs showed decreased proliferation and increased apoptosis (Zhang et al, 2018b). Those findings indicate multifunctional roles of NANOG in driving differentiation into PGCLCs.

The transcriptional profiles of Ck- and 4TF-induced PGCLCs resembled PGCs at different stages. This disparity partially arises from the essential role of WNT signaling and the mesodermal program in the Ck-induction system, whereas the 4TF-induction system circumvents these requirements. Combining insights from previous studies (Nakaki et al, 2013; Murakami et al, 2016), all PGCLC systems using TF overexpression (Nanog, 3TF, and 4TF) drive differentiation independently of the BMP4-WNT-T pathway. The gene regulatory axis of early-to-late PGC development established in 4TF-induced PGCLCs propels their differentiation beyond what Ck achieved within the same timeframe. Consequently, in the 4TF but not Ck-induced PGCLCs, the late PGC regulatory network was triggered as indicated by expression of genes involved in piRNA processes (such as *Piwil2/Mili*, *Tdrd5*, *Mov10l1*, *Tdrd7*, and *Tex19.2*). Although these findings indicate more advanced development in 4TF-induced PGCLCs, single-cell RNA sequencing will be required to provide detailed insights into their differentiation trajectories. Interestingly, 4TF-induced PGCLCs retain the capacity for conversion into pluripotent EGCs. These findings highlight the remarkable plasticity of the germline epigenome, though the differences in epigenetic reprogramming potential compared to endogenous late PGCs remain to be elucidated.

It was surprising that several meiosis genes were expressed in the 4TF-induced PGCLCs, including *Stra8*, *Sycp1/2/3*, *Meiob*, and *Smc1b*. One possible explanation is that although the engineered ESCs were male, some fraction of cells may have initiated a meiotic transcription program as would normally occur in females, absent sex determination signals that emanate from somatic cells in vivo. We also observed expression of *Mael*, involved in pachytene piRNA processes (namely LINE-element suppression) a gene expressed strongly in mouse spermatocytes and which is essential for normal spermatogenesis in mice (Soper et al, 2008). However, MAEL protein is present at substantial levels in fetal oocytes and primordial follicles in both humans and mice (Kim et al, 2014; Malki et al, 2014), where it is also involved in suppressing LINE-1 activity that contributes to fetal oocyte attrition. Regarding the mechanisms by which these meiosis genes are transcribed before clear evidence of meiocytes in the D6PGCLC cultures, network analysis gives a clue. During germ cell development, *Tfap2c* is activated by *Prdm1* and *Prdm14* (Saitou and Hayashi, 2021), directly targeting *Dmrt1* in PGCLCs (Schemmer et al, 2013; Magnúsdóttir et al, 2013). Human DMRT1 upregulates late PGC markers (such as DAZL and PIWIL2), while DAZL enhances DMRT1's impact by repressing pluripotency and promoting expression of late PGC (Irie et al, 2023) and meiosis genes. The beginning of the meiosis transcriptional program in these cultures may precede developmental or cellular hallmarks of meiosis.

At present, gonadal and testicular somatic cells are needed for further differentiation of male PGCLCs (Eguizabal et al, 2011; Ishikura et al, 2021, 2016). Seminiferous tubule-like structures can be constructed by aggregating Ck-induced PGCLCs with gonadal somatic cells under a gas-liquid interphase culture condition (Ishikura et al, 2021, 2016). Given that 4TF-induced PGCLCs exhibit more advanced development compared to Ck-induced PGCLCs, we opted for the readily accessible neonatal testicular somatic cells to facilitate PGCLC induction, along with key cytokines (i.e., GDNF and bFGF) crucial for spermatogonial proliferation. While no seminiferous tubule-like structures were observed, clusters of germ cells were formed and surrounded by somatic cells. Similar to previous observations (Ishikura et al, 2021, 2016), a subset of germ cells underwent further differentiation into cells with spermatogonium-like characteristics. However, most of the germ cell population had progressed beyond the PGC stage (70% were DDX4+ GCNA+) but did not transition to the DDX4+/PLZF+ spermatogonial-like state. Further studies to characterize dynamic changes of these germ cells in aggregates at the single-cell level would be helpful, as would exploration of different co-culture systems. Some germ cells displayed the early meiosis marker STRA8 after culture at 34 °C, but further work is necessary to explore whether these are truly meiotic cells that can complete meiosis and progress to later stages of spermatogenesis in vivo or in vitro.

Unraveling the genetic and epigenetic factors governing gametogenesis is crucial for understanding the causes of developmental or reproductive abnormalities. Though genomic analyses have characterized gene expression patterns and epigenetic modifications during germ cell development (Bleckwehl et al, 2021; Li et al, 2021; Alexander et al, 2023), the field has been hampered by the lack of high-throughput platforms for genome-wide functional studies. Massively parallel CRISPR-based screens require large number of cells to accommodate libraries containing tens of thousands of sgRNAs. Previously, a screen of 199 epigenetic modifiers was performed using individual RNAi knockdowns under standard conditions, i.e., exposure of individual EBs to cytokines in 96-well plates (Mochizuki et al, 2018). By contrast, the 4TF-inducible system enabled *en masse* gene knockdowns using Petri dishes and sequencing-based screening of 701 "epigenetic" genes for key roles in PGC development. Our study not only identified epigenetic events influencing the generation and development of PGCs, but also to draw links between epigenetic modifications and environmental factors (such as medications and dietary supplements) that might influence gametogenesis and possibly fertility.

The 4TF-induction system bypasses the need for BMP4 and WNT3, enabling direct programming of EpiLCs into PGCLCs. Consequently, our CRISPRi screen likely did not query epigenetic regulators involved during the period of BMP signaling and the activation of germline-specific TFs. As mentioned earlier, an RNAi-based epigenetic screen was described that utilized a Blimp1 reporter for PGCLC identification. *Blimp1* expression is equivalent to ~E6.25, and they likely preferentially identified epigenetic genes that regulate the transition of epiblast cells to the PGC fate, as well as the repression of somatic networks. In contrast, our platform preferentially queried epigenetic factors that regulate PGC differentiation downstream of germline TF expression. Disruption of these factors holds the potential to impede or override the germline transcription network.

The dietary supplement SB is beneficial for intestinal homeostasis and energy metabolism by enhancing intestinal barrier function and mucosal immunity with its anti-inflammatory properties (Liu et al, 2018). However, the potential effects of maternal SB supplementation upon fetal development are poorly explored. VPA is a widely prescribed drug used to treat epilepsy, bipolar disorder, and migraines. There is an elevated risk of birth defects (such as cleft lip, heart, and neural tube defects) in children born to women taking VPA during pregnancy (Jentink et al, 2010), but possible impacts on germ cell development remain unknown. Our results raise the possibility that fetal exposure to HDAC inhibitors can affect germ cell development, manifesting as atrophic seminiferous tubules and hypospermatogenesis in adulthood. However, further studies are needed to elucidate the mechanisms (specific roles of HDAC isoforms and other NCOR2 co-repressors), developmental stages, and cell types (e.g., PGCs, SSCs) that are impacted by HDAC inhibitors to yield the observed phenotypes. Although the HDACi-treated PGCLC induction cultures indeed showed increased H3K27ac levels (Fig. EV3H,I), it remains to be seen if the in vivo exposure caused the spermatogenesis phenotypes strictly via this epigenetic mechanism.

Interestingly, the ovarian reserve was unaffected even though treatment occurred before sex determination. It is possible that compromised cells were preferentially lost in the process of fetal oocyte attrition, which normally eliminates up to 80% of oocytes (Kurilo, 1981; Baker, 1963). Alternatively, the molecular impacts of treatments may only manifest in prospermatogonia or SSCs. Although the recommended clinical doses of SB and VPA are lower than those utilized in our study, we cannot say if human fetuses are more or less sensitive to these drugs than mouse embryos with respect to germline development. Our findings contribute to an expanding body of evidence indicating that epigenetic factors and dietary choices can impact male fertility (Babakhanzadeh et al, 2020; Mann et al, 2020).

In mice, competence for germline specification distinguishes the formative state of pluripotency from the naive and primed states (Smith, 2017). Studies using Ck-, 3TF- and Nanog-induction systems have demonstrated that a formative state is essential for robust induction of a PGC-like state (Nakaki et al, 2013; Hayashi et al, 2011; Murakami et al, 2016). Induction of 3TF in naive ESC resulted in a unique phenotype characterized by intense activation of the Stella reporter without forming a PGC-like state. The transcriptomes of these Stella+ cells were close to those of ESCs, but not PGCLCs/PGCs (Nakaki et al, 2013). Since Nanog abundantly expressed in naive ESC, the above result also indicates that induction of 4TFs in naive ESCs fails to form PGCLCs. Given that both formative ESCs and EpiLCs respond directly to PGC specification, it was unclear whether exposure of formative ESCs to 4TF overexpression was compatible with efficient differentiation into PGCLCs. Recent advancements have identified four distinct sub-stages of formative pluripotent stem cells, exhibiting functional characteristics and WNT/β-catenin signaling modulation (Luo et al, 2023; Yu et al, 2021; Kinoshita et al, 2021; Wang et al, 2021). In our study, we cultivated formative ESCs using one of these protocols (Yu et al, 2021), and revealed that 4TFs can drive both formative ESCs and EpiLCs into PGCLCs. PGCLCs (Stella-eGFP+ cells) differentiated from formative ESCs exhibit similar further developmental potential with those from EpiLCs. Importantly, enables simplification of the PGCLC induction process, as

formative ESCs can be stably maintained and would not require preliminary induction into the transient EpiLC-like state from naive ESCs.

Our CRISPRi screen identified 53 epigenetic candidates that compromised PGCLC formation, and those that most impaired PGCLC differentiation are normally upregulated in PGCLCs compared to naive ESCs and EpiLCs, including Rai1, Smarcd2, Kdm5d, Jak2, Ss18l1, and Ncor2. Knockdown of Ncor2 was shown to inhibit the formation of BLIMP1+ PGCLCs, suggesting an important role in mediating histone deacetylation in germ cell lineage specification (Mochizuki et al, 2018). As a component of the nuclear receptor corepressor complex, Ncor2 primarily contributes to gene repression through histone deacetylation. We confirmed Ncor2's impact on PGCLC development using siRNA to transiently suppress Ncor2, which led to the depletion of ITGB3+/Stella-eGFP+ PGCLCs. These results suggest that Ncor2 plays a crucial role in the suppression of transcriptional programs opposing the germline lineage fate. This gene suppression may be associated with the recruitment of Class I and Class IIa HDACs by the NCOR2 co-repressor complex. Previous work by Mochizuki et al demonstrated that BLIMP1-HDAC3-mediated somatic gene repression is vital for PGC fate determination. Additionally, they showed that Ncor2 knockdown caused BLIMP1+ PGCLCs to have elevated expression of somatic lineage genes such as Hoxb1, Hand1, Sox17, and Gata4 (Mochizuki et al, 2018). We did not identify individual Class I or Class IIa HDACs in our screen, leading us to speculate that Ncor2 inhibits somatic lineage development by interacting with Class I and Class IIa HDACs, and that there is redundancy amongst various HDACs.

The transcriptomes of Stella-eGFP+ and Stella-eGFP- cells following HDACi treatment fell into distinct clusters, supporting Stella as a reliable marker for mouse PGC development. Indeed, downregulated genes in Stella-eGFP- cells (vs. Stella-eGFP+) contained those related to reproduction, suggesting a suppression of cell fate. GSVA on reproductive pathways also indicated loss of germline fate even in the HDACi-treated Stella-eGFP+ PGCLCs, as indicated by relative depletion of PGC-associated transcripts such as Nanos3 and Dazl. Interestingly, unperturbed Stella-eGFP+ PGCLCs expressed several meiosis-related genes, and these meiotic genes were downregulated in the HDACi-treated cells. Even though there was no indication that the differentiated cells were differentiating into meiocytes, the PGCLCs may be partially inducing a meiotic transcriptional program that HDACi treatment disrupts. Additionally, HDACi-treated Stella-eGFP+ PGCLCs exhibited upregulation of pathways associated with mesoderm development, and there was significant upregulation of GO terms related to somatic lineage development in Stella-eGFP- cells. In mice, the induction Prdm1 and Prdm14 is crucial for driving PGC fate by suppressing mesoderm lineage genes (Saitou et al, 2002).

The coordinated regulation of BMP and WNT/β-Catenin signaling is pivotal for the specification of mouse PGCs from the proximal epiblast (Saitou and Hayashi, 2021). However, signaling through WNT/β-Catenin alone is insufficient for germline commitment. While the activation of WNT/β-Catenin does enhance the expression of T, it simultaneously triggers numerous mesoderm transcriptional regulators such as Sp5, Hoxa1, Hoxb1, Nkx1-2, Msx2, and Rreb1 (Aramaki et al, 2013). The exposure to BMP4 of cells in the proximal epiblast suppresses these mesoderm factors, creating an optimal environment for T to activate Prdm1

and *Prdm14*, initiating the emergence of PGCs. In HDACi-treated Stella-eGFP⁻ cells, we noted upregulation of these mesoderm regulators, but downregulation of *Ifitm3*, which is activated in PGCs by WNT3A and BMP4 signaling (Hayashi et al, 2007). This suggests suppression of germline fate in the presence of elevated signaling from other lineage differentiation factors. These findings support the idea that exposure to HDACi can negatively impact germline development.

Besides *Ncor2*, our screen identified several other candidate epigenetic regulators of PGC development. These include BLIMP1, which suppresses somatic gene expression (Ohinata et al, 2005), and TLE1, which interacts with the BLIMP1 repression domain to function as a co-repressor complex in B cells (Yu et al, 2000). Loss of *Tle1* is linked with elevated acetylation levels and gene activation (Ali et al, 2010), suggesting a role analogous to *Ncor2* in suppressing somatic gene expression in the germline. Other hits included Janus kinase 2 (JAK2), the absence of which causes a loss of gonadal PGCs in chicken (Zhang et al, 2017), and BMI1, an element of the Polycomb repressive complex 1 (PRC1) that is vital for the preservation of spermatogonial stem cells (Komai et al, 2014). However, its significance in prenatal germ cells has not been elucidated. Focused studies of these and other genes identified in the screen will be needed to validate and understand their roles.

This scalable and efficient 4TF-induction system offers a valuable tool to conduct unbiased genetic screens impacting PGCLC formation. Besides CRISPRi as described here, this platform enables other CRISPR-based perturbation screens, including knockouts, upregulation via CRISPR activation (CRISPRa), combinatorial disruptions, and single-cell transcriptomics as a phenotypic readout (Chavez et al, 2015; Dixit et al, 2016). Furthermore, a useful aspect of the 4TF system is that the derived PGCLCs persisted for prolonged periods in culture, enabling various studies involving, for example, large amounts of material for biochemical analyses, genetic manipulations initiated after cells reach this advanced stage, responses to drug or biologic exposures, or as a starting point for exploring further differentiation. Finally, considering that genetic causes underlie ~50% of human infertility (Ding and Schimenti, 2021), parallelized screening of genetic variants, particularly rare or minor alleles that are classified as variants of uncertain significance (VUS), is another potential application. Functionally interpreting the VUS within genes that are essential for gametogenesis, along with achieving a comprehensive understanding of epigenetic networks, are important for genetic counseling, clinical applications, and potentially gene correction therapies for individuals to produce viable gametes in vitro *or* in vivo (Ding and Schimenti, 2021; Ding et al, 2023; Ding and Schimenti, 2023).

# Methods

**Reagent and tools table**

| Reagent/resource | Reference or source | Identifier or Cat# |
|---|---|---|
| **Experimental models** | | |
| C57BL/6J-Kit^Wv/J | The Jackson Laboratory | #000049 |
| WB/ReJ Kit^W/J | The Jackson Laboratory | #000692 |
| B6(Cg)-Tyr^c-2J/J | The Jackson Laboratory | #000058 |
| C57BL/6 J | The Jackson Laboratory | #000664 |
| HEK293T | ATCC | CRL-3216 |
| Stella-eGFP ESC | Prof. M. Azim Surani | N/A |
| **Oligonucleotides and sequence-based reagents** | | |
| Primers | Experiment | Sequence (5′ to 3′) |
| rtTA geno F | Genotyping | ATGTCTAGACTGGACAAGAGC |
| rtTA geno R | Genotyping | CAGAAAGTCTTGCCATGAC |
| Tfap2c geno F | Genotyping | GTGAAAGTCGAGCTCGGTACCC |
| Tfap2c geno R | Genotyping | CCAGATGCGAGTAATGGTCGG |
| Common F | Genotyping | CTCCTGGGCAACGTGCTG |
| Prdm1 geno R | Genotyping | CCAGTCTCTGCCAGTCCTTG |
| Prdm14 geno R | Genotyping | GTGGTGGCGAGGTTCCTAA |
| Nanog geno R | Genotyping | CATCTTCTGCTTCCTGGCAAGGA |
| H11 outside R | Genotyping | ctacactcctcccacccagttg |
| H11 inside F | Genotyping | CCTCCCCCTGAACCTGAAACAT |
| Prdm1 pF | Build the vectors | CTCGAGGCCACCatgttggatcttctcttggaaaaacgtgtgg |
| Prdm1 pR | Build the vectors | GCGGCCGCttaaggatccatcggttcaactgtttcttgtttgac |
| Prdm14 pF | Build the vectors | CTCGAGGCCACCatggccttaccgc |
| Prdm14 pR | Build the vectors | GCGGCCGCctagcaggttttatgaagcctcatg |

| Reagent/resource | Reference or source | Identifier or Cat# |
|---|---|---|
| Snrpn F | Bisulfite sequencing | TATGTAATATGATATAGTTTAGAAATTAG |
| Snrpn R | Bisulfite sequencing | AATAAACCCAAATCTAAAATATTTTAATC |
| Snrpn nest F | Bisulfite sequencing | AATTTGTGTGATGTTTGTAATTATTTGG |
| Snrpn nest R | Bisulfite sequencing | ATAAAATACACTTTCACTACTAAAATCC |
| H19 F | Bisulfite sequencing | GAGTATTTAGGAGGTATAAGAATT |
| H19 R | Bisulfite sequencing | ATCAAAAACTAACATAAACCCCT |
| H19 nest F | Bisulfite sequencing | GTAAGGAGATTATGTTTATTTTTGG |
| H19 nest R | Bisulfite sequencing | CCTCATTAATCCCATAACTAT |
| Dnmt3a qF | qPCR | GACTCGCGTGCAATAACCTTAG |
| Dnmt3a qR | qPCR | GGTCACTTTCCCTCACTCTGG |
| Dnmt3b qF | qPCR | CTCGCAAGGTGTGGGCTTTTGTAAC |
| Dnmt3b qR | qPCR | CTGGGCATCTGTCATCTTTGCACC |
| Ncor2 qF | qPCR | GATGACCCCATGAAGGTCTACA |
| Ncor2 qR | qPCR | GGCCAAAGTTCTTAGGGTGCT |
| Gapdh qF | qPCR | CTTTGTCAAGCTCATTTCCTGG |
| Gapdh qR | qPCR | TCTTGCTCAGTGTCCTTGC |
| Miseq_Adaptor F | MiSeq | AATGATACGGCGACCACCGAG ATCTACACACACTCTTTCCCTA CACGACGCTCTTCCGATCTTAT CTTGTGGAAAGGACGAAACACCG |
| Miseq_Adaptor R | MiSeq | CAAGCAGAAGACGGCATACGAGAT GTGACTGGAGTTCAGACGTGTGCT CTTCCGATCTCTTTAGTTTGTATGT CTGTTGCTATTATGTCTACTATTCTTTCC |
| seqRead Miseq | MiSeq | GACGCTCTTCCGATCTTATCTTGTGGAA AGGACGAAACACCG |
| CRISPRi F | CRISPRi cloning | AGGCACTTGCTCGTACGACG |
| CRISPRi R | CRISPRi cloning | TTAAGGTGCCGGGCCCACAT |
| dCAS9 F | dCas9-KRAB expressing | CAAGGTGCTGGGCAACACCGA |
| dCAS9 R | dCas9-KRAB expressing | TCGTCCACCTTGGCCATCTCG |
| Nanog RT F | Reverse transcription PCR | ACCTGAGCTATAAGCAGGTTAAGAC |
| Prdm1 RT F | Reverse transcription PCR | gtgacctccatggtggagaag |
| Prdm14 RT F | Reverse transcription PCR | ggagagaagcccttcaagtgc |
| bGH pA R | Reverse transcription PCR | GGGGGAGGGGCAAACAACA |
| Tfap2c RT F | Reverse transcription PCR | GGGGACGGAATGAGATGGC |
| WPRE R | Reverse transcription PCR | AAGGCATTAAAGCAGCGTATCC |
| rtTAM2 F | Reverse transcription PCR | ATGTCTAGACTGGACAAGAGC |
| rtTAM2 R | Reverse transcription PCR | CAGAAAGTCTTGCCATGAC |
| **Antibodies** | | |
| DDX4 (Rb) | Abcam | ab13840 |
| DDX4 (Ms) | Abcam | ab27591 |
| PLZF (Ms) | Active motif | 39987 |
| GCNA (Rat) | Abcam | ab82527 |
| DAZL (Rb) | ThermoFisher | PA5-116570 |
| NANOS3 (Rb) | Abcam | ab70001 |
| STRA8 (Rb) | Abcam | ab49602 |
| SOX9 (Rb) | Millipore | AB5535 |
| ZO1 (Rb) | ThermoFisher | 40-2200 |

| Reagent/resource | Reference or source | Identifier or Cat# |
|---|---|---|
| E-Cadherin (Ms) | TaKaRa | M108 |
| SSEA1 (Ms) | Millipore | MAB4301 |
| SOX2 (Ms) | ThermoFisher | MA1-014 |
| H3K9me3 (Rb) | Upstate | 07-442 |
| H3K27me3 (Rb) | EPIGENTEK | A-4039-025 |
| Goat anti-Mouse-488 | Invitrogen | A11001 |
| Goat anti-Rabbit-594 | Invitrogen | A11012 |
| Goat anti-Rat-594 | Invitrogen | A11007 |
| Beta-actin (Ms) | Sigma | 1978 |
| CAS9 (Ms) | Abcam | ab191468 |
| H3K27ac (Rb) | Abcam | ab4729 |
| H3 (Rb) | Abcam | ab1791 |
| Goat anti-Mouse IgG, HRP | ThermoFisher | 31430 |
| Anti-rabbit IgG, HRP | Cell signaling | 7074S |
| PE (Phycoerythrin) mouse/rat anti-CD61 | Invitrogen | 12-0611-82 |
| Alexa Fluor® 647 mouse/human anti-SSEA1 | Biolegend | 125608 |
| **Recombinant DNA** | | |
| pPBhCMV1-Prdm1-pA | This study | N/A |
| pPBhCMV1-Prdm14-pA | This study | N/A |
| pPBhCMV1-Nanog-pA | Prof. M. Azim Surani | N/A |
| PCR4-Shh::lacZ-H11 | Addgene | #139098 |
| pPyCAG-Pbase | Prof. M. Azim Surani | N/A |
| pPB-CAG-rtTA-IN | Addgene | #60612 |
| pSpCas9(BB)-2A-PuroR | Addgene | #62988 |
| psPAX2 | Addgene | #12260 |
| pMD2.G | Addgene | #12259 |
| pLV-tetO-Tfap2c | Addgene | #70269 |
| pLX_311-KRAB-dCas9 | Addgene | #96918 |
| pGEM-T Easy vector | Promega | A1360 |
| pXPR_050 | Addgene | #96925 |
| **Chemicals and other reagents** | | |
| TransIT®-LT1 Transfection Reagent | Mirus Bio™ | MIR2304 |
| Geneticin™ | Life Technologies | 10131035 |
| Zeocin® | InvivoGen | ant-zn-05 |
| LIF | Peprotech | 300-05 |
| PD0325901 | Reprocell | 04-0006-02 |
| CHIR99021 | Reprocell | 04-0004-02 |
| poly-L-ornithine | Sigma-Aldrich | P3655 |
| laminin | Corning | 354232 |
| bovine plasma fibronectin | Sigma-Aldrich | F1141 |
| Activin A | Peprotech | 120-14E |
| bFGF | Gibco | 13256-029 |
| KSR | ThermoFisher | 10828010 |
| Dox | Sigma | D9891 |
| Propidium iodide | Invitrogen | P3566 |

| Reagent/resource | Reference or source | Identifier or Cat# |
|---|---|---|
| Goat serum | Sigma | NS02L |
| GDNF | PeproTech | 450-44 |
| Insulin-Transferrin-Selenium | ThermoFisher Scientific | 41400045 |
| MEMα | ThermoFisher Scientific | 12571063 |
| DeadEnd™ Fluorometric TUNEL System | Promega | G3250 |
| Vector Red Alkaline Phosphatase Substrate Kit | Vector Laboratories | SK-5100 |
| β-Galactosidase Staining Kit | Cell Signaling Technology | #9860 |
| Vectashield antifade mounting medium | Vector | H-1000 |
| EZ DNA methylation-lightning kits | ZYMO Research | D5030 |
| E.Z.N.A.® Total RNA Kit I | Omega BIO-TEK | R6834 |
| TRIzol® reagent | ThermoFisher Scientific | 15596018 |
| qScript™ cDNA SuperMix kit | Quantabio | 95048 |
| RT² SYBR® Green qPCR Mastermixes | Qiagen | 33050 |
| T7 ligase | NEB | M0318S |
| Bsmb1 | NEB | R0580 |
| Stbl4 competent cells | Invitrogen | 11635018 |
| Blasticidin | Gibco | A1113903 |
| Puromycin | Gibco | A1113803 |
| Genomic DNA Purification Kit | Thermo Scientific | K0722 |
| Q5 High-Fidelity DNA Polymerase | NEB | M0491L |
| Gel Extraction Kit | Qiagen | 28704 |
| control siNT | Dharmacon | D-001810-10-05 |
| siNcor2 | Dharmacon | L-045364-00-0005 |
| jetPRIME transfection reagent | Polyplus | 101000046 |
| Sodium butyrate | Sigma | B5887 |
| valproic acid | Sigma | P4543 |
| protease inhibitors | Sigma | 04693159001 |
| RIPA buffer | ThermoFisher Scientific | 89900 |
| Phosphatase inhibitors | Sigma | 4906845001 |
| BCA Protein Assay Kit | Pierce | 23250 |
| Gradient polyacrylamide gel | Bio-Rad | 4561083EDU |
| PVDF membrane | Millipore | IPVH00010 |
| HRR substrate | Millipore | WBLUC0500 |
| NEBNext Ultra II Directional RNA Library Prep Kit | NEB | E7765 |
| **Software** | | |
| Epifactors | https://epifactors.autosome.org/ | |
| GPP sgRNA Design tool | https://portals.broadinstitute.org/gpp/public/analysis-tools/sgrna-design | |
| ScreenProcessing Tools | https://github.com/mhorlbeck/ScreenProcessing | |
| MAGeCK, MAGeCK-VISPR, and MAGeCKFlute Tools | https://github.com/WubingZhang/MAGeCKFlute | |
| DESeq2 | https://bioconductor.org/packages/release/bioc/html/DESeq2.html | |
| SARTools | https://github.com/PF2-pasteur-fr/SARTools | |
| RRHO2 | https://github.com/RRHO2/RRHO2 | |

| Reagent/resource | Reference or source | Identifier or Cat# |
|---|---|---|
| GSVA | https://alexslemonade.github.io/refinebio-examples/ | |
| GSEA | https://www.gsea-msigdb.org/gsea/index.jsp | |
| iDEP 1.1 | https://bioinformatics.sdstate.edu/idep11/ | |
| g:Profiler | https://biit.cs.ut.ee/gprofiler/gost | |
| ShinyGo 0.80 | http://bioinformatics.sdstate.edu/go80/ | |
| ggplot2 | https://ggplot2.tidyverse.org | |
| STRING 12.0 | https://string-db.org | |
| FlowJo | https://www.flowjo.com | |
| BioRender | www.biorender.com | |
| Cytoscape | https://cytoscape.org | |
| EnrichmentMap | https://apps.cytoscape.org/apps/enrichmentmap | |

## Mice and cell lines

All animal usage was approved by Cornell University's Institutional Animal Care and Use Committee, under protocol 2004-0038 to J.C.S. Four- to 8-day-old Kit$^{W/Wv}$ male mice generated by crossing C57BL/6J-Kit$^{Wv}$/J and WB/ReJ Kit$^W$/J were used in this study. C57BL/6J-Kit$^{Wv}$/J (#000049), WB/ReJ Kit$^W$/J (#000692), B6(Cg)-Tyr$^{c-2J}$/J (#000058) and C57BL/6J (#000664) mice were purchased from The Jackson Laboratory. The Stella-eGFP transgenic ESC line (Stella-eGFP ESCs) with Stella flanking sequences coupled to eGFP as the reporter was kindly provided by Prof. M. Azim Surani (Payer et al, 2006).

## Vectors

To construct pPBhCMV1-Prdm1-pA and pPBhCMV1-Prdm14-pA, mouse *Prdm1* and *Prdm14* coding sequences were cloned into pPBhCMV1-Nanog-pA (a gift from M. Azim Surani) by PCR flanked with *Xho*I and *Not*I restriction sites. To construct targeting vectors containing enhancer-reporter transgenes, PCR-amplified sequences of mouse *Prdm1* and *Prdm14* enhancers (Murakami et al, 2016) bearing terminal *Not*I restriction sites were cloned into PCR4-*Shh::lacZ*-H11 (Addgene, #139098). Plasmid clones were validated by Sanger sequencing. pPyCAG-Pbase and pPB-CAG-rtTA-IN (Addgene, #60612) were kindly provided by M. Azim Surani. pSpCas9(BB)-2A-PuroR (px459) V2.0 (Addgene, #62988), psPAX2 (Addgene, #12260), pMD2.G (Addgene, #12259), pLV-tetO-Tfap2c (Addgene, #70269), pLX_311-KRAB-dCas9 (Addgene, #96918) and pXPR_050 (Addgene, #96925) were obtained from Addgene. Primer sequences are listed in the "Reagents and tools table".

## Microinjection for chimera production

The ESCs were microinjected to B6(Cg)-Tyr$^{c-2J}$/J blastocysts by standard methods in Cornell's Transgenic Core Facility (Luo et al, 2014).

## Derivation of genetically modified ESC clones

A *piggyBac* transposon expression system was used to stably integrate single copies of various vectors that were doxycycline (Tet) inducible.

To establish the Tet-on expression system, pPBhCMV1-Prdm1-pA, pPBhCMV1-Prdm14-pA, and pPBhCMV1-Nanog-pA vectors were transfected using TransIT®-LT1 Transfection Reagent (Mirus Bio™, MIR2304) into ESCs together with pPBCAG-rtTA and pPyCAG-PBase vectors. After 1 week of Geneticin™ (80 µg/ml; Life Technologies, 10131035) selection (all vectors except pPyCAG-PBase contain neomycin resistance genes), clones containing desired inserts were identified. Cells infected with pLV-tetO-Tfap2c lentivirus (described below) were selected by Zeocin® (InvivoGen, ant-zn-05) to identify transformants.

To construct ESC lines containing enhancer-reporters, the targeting vector PCR4-*Shh::lacZ*-H11 and H11 gRNA-expression plasmid px459 were transfected into cells bearing Tet-inducible *Nanog* alone. After puromycin selection for 48 h, single colonies were expanded, and correctly targeted clones were identified using primers listed in the "Reagents and tools table".

## ESC, iPSC culture, and PGCLC induction

Mouse ESCs were maintained on γ-irradiated feeder MEFs (mouse embryonic fibroblast cells) in DMEM containing 15% FBS, penicillin–streptomycin, NEAA, β-mercaptoethanol, 1000 U/ml LIF (Peprotech, 300-05). Naive ESCs were cultured in 2i+ LIF medium containing 1×N2B27, 1 µM PD0325901 (Reprocell, 04-0006-02), 3 µM CHIR99021 (Reprocell, 04-0004-02), and 1000 U/ml LIF on wells coated with 0.01% poly-L-ornithine (Sigma-Aldrich, P3655) and 10 ng/ml laminin (Corning, 354232).

For the formative state conversion, naive ESCs and iPSCs were dissociated into single cells and plated on MEFs plate in 2i +LIF medium. Follow the protocol described previously (Yu et al, 2021), the next day, medium was changed to formative medium (N2B27 medium containing 10 ng/ml bFGF, 10 ng/ml Activin A and 3 µM CHIR99021) for further culture. It usually takes about 2–4 passages for full conversion. Formative ESCs and iPSCs were maintained in formative medium.

For PGCLC induction, EpiLCs were induced from naive ESCs as described (Hayashi and Saitou, 2013), with details as follows:

- Before EpiLC differentiation, coat a 12-well plate with plasma fibronectin (Sigma-Aldrich, F1141) as follows: dilute 10 µL of fibronectin with 600 µL of PBS to obtain a 16.6 µL/mL fibronectin solution, pipette the solution into one well of a 12-well plate,

incubate for at least 1 h in a tissue culture incubator, and then aspirate the fibronectin solution just before plating the cells.

- To passage cells, aspirate medium, wash the cells once with PBS, and add 1 mL of Trypsin/EDTA. Incubate the plate for 3 min at 37 °C.
- Inactivate Trypsin by adding 1 mL of MEF medium, then and pipette up and down with a P1000 pipettor to dissociate the colonies into single cells.
- Pellet the cell suspension in a 15 mL Falcon tube at $220 \times g$ for 3 min and remove the supernatant.
- Resuspend the cells with EpiLC medium (N2B27 medium containing 20 ng/mL Activin A (Peprotech, 120-14E), 12 ng/mL bFGF (Gibco, 13256-029), and 1% knockout serum replacement (KSR, ThermoFisher, 10828010).
- Transfer $1 \times 10^5$ cells to the fibronectin-coated 12-well plate.
- Replace EpiLC medium the next day.
- A total of 48 h after plating, dissociate the EpiLCs and count the cells.
- For standard PGCLC differentiation, mix thoroughly $2 \times 10^5$ cells with 10 mL Gk15 medium (GMEM with 15% KSR, 0.1 mM NEAA, 1 mM sodium pyruvate, 0.1 mM β-mercaptoethanol, penicillin–streptomycin, and 2 mM L-glutamine) and 2 µg/mL Dox in a reservoir. Using a multichannel pipet, transfer 100 µL of the cell suspension in each well of the U-bottom 96-well plate (ThermoFisher Scientific, 268200). Gently pipet up and down in the reservoir between each transfer. To scale up PGCLC differentiation, transfer $8 \times 10^4$ cells in 1 mL Gk15 medium with Dox in each well of an untreated 12-well plate (ThermoFisher Scientific, 150200). Alternatively, transfer $1 \times 10^6$ cells in 8 mL Gk15 medium with Dox into a sterile, non-tissue culture-treated bacteriological 100-mm Petri dish.
- To induce PGCLCs from formative ESCs, after removing the feeder cells, $2 \times 10^3$ cells per well were seeded in the U-bottom 96-well plate. Alternatively, $8 \times 10^4$ cells per well were seeded in an untreated 12-well plate, or $1 \times 10^6$ cells were seeded into a 100-mm Petri dish in GK15 medium. To sustain the PGCLCs, day 6 EBs in a 12-well plate were dissociated into single cells, and $8 \times 10^4$ cells per well were seeded in an untreated 12-well plate. The cells were passaged every 4 days. Transgenes were induced by the addition of 2 µg/ml Dox at day 0 of PGCLC induction.

## Flow cytometry

To flow sort PGCLCs, single-cell suspensions were stained with 1 µg/ml propidium iodide (PI, Invitrogen, P3566) and filtered through a 70-µm cell strainer. eGFP⁺ cells were sorted using a BD FACSMelody 4-Way Sorter. To analyze the PGCLCs, suspended cells were washed with PBS and fixed in 70% ethanol. For staining, cells were washed in MACS buffer (1×PBS, 0.5%BSA, and 2 mM EDTA) and blocked with 5% goat serum (Sigma, NS02L) in MACS buffer at 4 °C for 1 h. Cells were then stained with PE (Phycoerythrin) mouse/rat anti-CD61 and Alexa Fluor® 647 mouse/human anti-SSEA1 antibodies (reagents and tools table) at 4 °C for 1 h. Cells were then washed in MACS buffer twice and filtered through a 35-µm cell strainer (Falcon, 352235). The stained cells were analyzed using a BD FACSymphony A3 cytometer. FACS data were analyzed using FlowJo™ v10.4 or the flowCore package in R.

## Cell aggregate cultures for further differentiation

Sorted D6PGCLCs were mixed with dissociated neonatal testicular cells from Kit^{W/Wv} mice with 10:1 ratio and co-cultured for 2 days in a U-bottom 96-well using GK15 medium. The aggregates were subsequently transferred to a trans-well plate in N2B27 medium containing 20 ng/ml bFGF, 40 ng/ml GDNF (PeproTech, 450-44), 1000 U/ml LIF, 1× Insulin-Transferrin-Selenium (ThermoFisher Scientific, 41400045) and 15% KSR for an additional 2 weeks. The aggregates were cultured at 37 °C in a 5% $CO_2$-95% air atmosphere. Two weeks later, the aggregates were cultured on agarose gel stand in MEMα (Minimum Essential Medium α; ThermoFisher Scientific, 12571063) containing 15% KSR for an additional week, as described previously (Sato et al, 2011). The aggregates in this stage were cultured at 34 °C in a 5% $CO_2$–95% air atmosphere.

## Cell staining

ALP staining was performed using the Vector Red Alkaline Phosphatase Substrate Kit (Vector Laboratories, Burlingame, CA, USA) according to the manufacturer's directions. *LacZ* staining was performed using the Senescence β-Galactosidase Staining Kit (Cell Signaling Technology, #9860) according to the manufacturer's directions. For TUNEL staining, sections were deparaffinized and performed TUNEL staining using the DeadEnd™ Fluorometric TUNEL System (Promega; G3250) following the manufacturer's instructions.

For cell IF staining, the cells growing on cover slides were washed with PBS, fixed in 4% paraformaldehyde (PFA) for 10 min at RT, washed twice with PBS. For section IF staining, cell aggregates were fixed in 4% PFA and embedded in paraffin. The 6 µm sections were deparaffinized and performed antigen retrieval using sodium citrate buffer. The slides were incubated for 10 min at 37 °C in blocking buffer (PBS containing 10% normal goat serum). Then, they were incubated overnight in a humidified chamber at 4 °C with the antibodies listed in the "Reagents and tools table". After washing twice with PBS, the slides were incubated at 37 °C for 30 min with a 1:500 dilution of secondary antibodies (reagents and tools table), then incubated at 37 °C for 5 min with 500 ng/mL of DAPI and mounted using Vectashield antifade mounting medium (Vector, H-1000).

## Bisulfite sequencing

Methylation patterns in the differentially methylated domains (DMDs) of the paternally imprinted gene *H19* and maternally imprinted gene *Snrpn* were determined using bisulfite genomic sequencing as previously described (Zhou et al, 2016). Genomic DNA was isolated and bisulfite treatment was conducted with an EZ DNA methylation-lightning kits (ZYMO Research, D5030) according to the manufacturer's protocol. Bisulfate-converted DNA was subjected to PCR amplification of the *H19* and *Snrpn* DMDs. The primer sequences and PCR conditions for amplification are previously described and listed in the "Reagents and tools table" (Zhou et al, 2016). PCR products were sub-cloned into the pGEM-T Easy vector (Promega, A1360) and sequenced. Sequences were determined and analyzed using BiQ Analyzer software (Bock et al, 2005).

## Reverse transcription and qRT-PCR

Total RNA extraction from cells was performed by E.Z.N.A.® Total RNA Kit I (Omega BIO-TEK) according to the manufacturer's instructions or extracted with TRIzol® reagent (ThermoFisher Scientific, 15596018). Reverse transcription was performed using a qScript™ cDNA SuperMix kit (Quantabio, 95048). qRT-PCR was performed with a C1000 Touch™ Thermal Cycler (Bio-Rad) amplification system using RT² SYBR® Green qPCR Mastermixes (Qiagen) and primer sets specific for each gene ("Reagents and tools table"). The relative levels of transcripts were calculated using the $2^{-\Delta\Delta C_T}$ method, and gene expression levels were normalized to *Gapdh*.

## CRISPRi library design and gRNA cloning

Seven hundred and one epigenetic factors were selected from Epifactors, and 10 unique sgRNAs were designed for each gene using the GPP sgRNA Design tool. The CRISPRi library sgRNA oligos with *Bsm*b1 overhangs were synthesized by GenScript (Piscataway, NJ) and cloned into the lentiviral vector *pXPR_050* by Golden Gate cloning following published protocols (Read et al, 2017). Briefly, oligo pools were PCR-amplified using primers listed in the "Reagents and tools table". *pXPR_050* plasmids were digested using the Bsmb1 restriction enzyme (NEB, R0580). The synthesized double-strand oligo pools and digested *pXPR_050* plasmids were ligated using T7 ligase (NEB, M0318S). The *pXPR_050* plasmids were then transformed into Stbl4 competent cells (Invitrogen, 11635018). The transformed bacteria were then plated on LB plates with ampicillin resistance and grown for 24 h at 30 °C. To ensure high coverage of the CRISPRi library, bacterial colonies over 100× of the CRISPRi oligo library (~ 800,000) were collected and grown in SOC medium (Sigma, S1797). Plasmid DNA was isolated using a midi-prep kit (Invitrogen, K210005). The integrity of the CRISPRi library was validated by Illumina Miseq through Cornell Genomics Facility (detailed library preparation steps are presented below in "Library validation and CRISPRi dropout screening").

For CRISPR-assisted insertion of PCR4-*Shh::lacZ*-H11 vector into the H11 locus, we used a gRNA that was previously designed for this purpose (Kvon I 2020). Synthesized oligonucleotides were annealed and cloned into px459 vector. The sequences of primers are listed in the "Reagents and tools table".

## Lentiviral packaging and transduction

Lentiviral particles were produced by transfecting HEK293T cells with the following plasmids and amounts: 2.5 µg pMD2.G, 7.5 µg psPAX2, and 10 µg vectors (for *pLX_311* lentiviral expression of *dCas9-KRAB*, the entire constructed *pXPR_050* CRISPRi sgRNA expression library, and pLV-tetO-Tfap2c) using TransIT-LT1 (Mirus, MIR2305). HEK293T culture media was collected at both 48 and 72 h post transfection and concentrated using Amicon Ultra-15 columns (Millipore, UFC903024). ESCs were first transduced with lentiviral packaged *pLX_311* and selected with Blasticidin (Gibco, A1113903) to generate stable *dCas9-KRAB* expressing cell lines, which were validated by PCR (reagents and tools table) and Western blotting as described below. Approximately $1.6 \times 10^7$ ESCs with stable *dCas9-KRAB* expression were then transduced with the *pXPR_050* CRISPRi sgRNA expression library at a multiplicity of infection of 0.3 to ensure at least 500× coverage of the library. The cells were selected in puromycin (Gibco, A1113803) (1.2 mg/ml) for 2 days. All cells were infected with lentivirus for 48 h.

## Library validation and CRISPRi dropout screening

Genomic DNAs from cells carrying CRISPRi sgRNA library were extracted using a Genomic DNA Purification Kit (Thermo Scientific, K0722). To recover sgRNA sequences, both CRISPRi plasmid pools and genomic DNAs were PCR-amplified with Miseq_Adaptor primers (reagents and tools table) using Q5 High-Fidelity DNA Polymerase (NEB, M0491L). PCR samples were purified from 2% agarose gels using a Gel Extraction Kit (Qiagen, 28704), then sequenced on an Illumina Miseq (Cornell Genomics Facility) using the seqRead_Miseq primer ("Reagents and tools table"). To validate the integrity of the CRISPRi library, the raw counts were mapped to the original CRISPRi library using ScreenProcessing Tools (Liu et al, 2021). To identify CRISPRi dropout candidates, sgRNA representations from ESCs, *Stella-eGFP*+, and *Stella-eGFP*- cells were analyzed using MAGeCK, MAGeCK-VISPR, and MAGeCKFlute Tools (Wang et al, 2019).

## Knockdown and drug treatment during PGCLC induction

Twenty-four hours before PGCLC induction, EpiLCs were transfected with control siNT (ON-TARGETplus Non-targeting Control siRNA, Dharmacon, D-001810-10-05) and siNcor2 (ON-TARGETplus siRNA SMARTPool, Dharmacon, L-045364-00-0005) short dsRNA molecules using jetPRIME transfection reagent (Polyplus, 101000046) at a final concentration of 50 nM. EpiLCs were then differentiated into PGCLCs over 6 days. Knockdown efficiencies were measured at day 2. Sodium butyrate (Sigma; B5887) and valproic acid (Sigma, P4543) were dissolved in water at stock concentrations of 100 mg/ml and 1 M, respectively. To treat cells, dilutions were added to the PGCLC culture media at the start of PGCLC induction. Cells were collected 2, 4, or 6 days later. To prepare the cells for RNA-seq, the EBs were treated with 0.05 mg/ml SB or 0.5 mM VPA.

## Western blotting

Cells were lysed and homogenized using RIPA buffer (Thermo-Fisher Scientific, 89900) with protease inhibitors (Sigma, 04693159001) and phosphatase inhibitors (Sigma, 4906845001). The samples were centrifuged at 13,000 rpm (~ 15,000 G), and supernatants were used for western blot analysis. The protein concentrations were measured using a BCA Protein Assay Kit (Pierce, 23250). Samples were run through a gradient polyacrylamide gel (Bio-Rad, 4561083EDU) and transferred to PVDF membranes (Millipore, IPVH00010). Membranes were blocked by 5% nonfat milk for 1 h. The antibodies used for western blotting were listed in the "Reagents and tools table". The HRR substrate (Millipore, WBLUC0500) and ChemiDoc Imaging System (Bio-Rad) were used for protein visualization.

## Mouse treatment and reproductive phenotyping

For administration of drugs during pregnancy, E0.5 is the morning in which a vaginal plug was detected. Pregnant females ($N = 5$ for

each group) received daily injections of water, SB (600 mg/kg and 800 mg/kg), or VPA (300 mg/kg) from E5.5 to E11.5. Histology analysis, sperm and follicle counts were performed as described previously (Ding et al, 2020, 2023). To prepare the paraffin blocks, tissues were fixed overnight in Bouin's solution or 4% PFA at room temperature and were embedded by the Histology Lab at Cornell University. Paraffin sections at 6μm thick were deparaffinized and stained with hematoxylin and eosin. One cauda epididymis per male was used for sperm counting. To quantify follicle number, the ovary block was serially sectioned and stained by H&E. Every fifth section was scored for follicles at various stages.

## RNA-seq and data analysis

Total RNAs were purified from cells using TRIzol® reagent (Invitrogen, 15596018). Purified RNAs were used for Illumina RNA-seq library preparation with NEBNext Ultra II Directional RNA Library Prep Kit (NEB, E7765), and a minimum of 20 million raw reads were obtained. The raw counts were then aligned to mouse mm10 genome with ENSEMBL gene annotations. Differential analysis of RNA-seq data was done using the R package DESeq2. RRHO2 analysis was also performed using an R package (Cahill et al, 2018). k-Means clustering analysis was performed using iDEP 1.12 (Ge et al, 2018). GSEA was performed to explore enriched pathways and interpret RNA-seq data using predefined gene sets from the Molecular Signatures Database (v.7.1) in R packages (Subramanian et al, 2005). GSVA was carried out using the GSVA package in R (Hänzelmann et al, 2013). Gene sets enriched in the GSEA were organized into a graphical network produced using EnrichmentMap (Merico et al, 2010) plugins in Cytoscape (v.3.10.1) (Shannon et al, 2003), as described previously (Reimand et al, 2019). The online STRING database (version 12.0) was used for the generation of protein interaction (Szklarczyk et al, 2023). The upregulated DEGs ($log2FC > 2$, $P < 0.05$ and count > 250) from PGCLCs on days 2, 4, and 6 compared to ESC were integrated for the network analysis.

## Accession numbers of publicly available datasets from other depositors

The RNA-seq data of ESCs/EpiLCs/Ck_D2/4/6PGCLCs (GSE67259) and in vivo germ cells (GSE74094 and GSE87644) were downloaded from the GEO database. The ESCs/EpiLCs/3TF_D2/4PGCLCs microarray dataset is accession # GSE43775. The GEO number for RNA-seq data of formative ESCs is GSE135989. The GEO number for the microarray dataset of ESCs/Nanog_D4PGCLC/Ck_D4PGCLC is GSE71933. The GEO number for RNA-seq datasets of 6, 12, 24, and 48 h EpiLCs is GSE256381.

## Image processing and data statistics

Fluorescent images were captured by an Olympus XM10 camera. Bright-field images were captured by the Olympus SC30 camera. Cropping, color, and contrast adjustments were made with Adobe Photoshop CC 2024, using identical background adjustments for all images. Fluorescence intensity was measured by ImageJ. All data were expressed as mean ± SEM. Statistical tests were carried out using a paired Student's $t$ test or one-way analysis of variance (ANOVA) followed by the Tukey's post hoc test with R. Graph

generation was performed using R. The schematic diagrams were created using BioRender.

## Data availability

The RNA-seq datasets produced in this study in Gene Expression Omnibus: GSE256104, GSE259289, GSE259290.

The source data of this paper are collected in the following database record: biostudies:S-SCDT-10_1038-S44319-025-00633-z.

## Peer review information

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

## Acknowledgements

The authors would like to thank R Munroe and C Abratte of Cornell's transgenic facility for generating the mice, A Surani for Stella-eGFP ESCs and several plasmids, and JC Bloom for OG2 iPSCs. This work is supported by a grant from the National Institutes of Health (R01 HD082568 to JCS), a Center grant (1P50HD096723) from the National Institute of Child Health and Human Development (K Orwig: PI; JCS: Lead Project 2), and contract CO29155 from the NY State Stem Cell Program (NYSTEM). XD was supported by a postdoctoral fellowship from the Empire State Stem Cell Fund through New York State Department of Health contract no. C30293GG.

## Author contributions

**Liangdao Li**: Conceptualization; Formal analysis; Investigation; Methodology; Writing—original draft; Writing—review and editing. **Jingyi Gao**: Data curation; Formal analysis. **Dain Yi**: Investigation. **Alex P Sheft**: Investigation. **John C Schimenti**: Conceptualization; Supervision; Funding acquisition; Writing—original draft; Project administration; Writing—review and editing. **Xinbao Ding**: Conceptualization; Formal analysis; Investigation; Writing—original draft; Writing—review and editing.

Source data underlying figure panels in this paper may have individual authorship assigned. Where available, figure panel/source data authorship is listed in the following database record: biostudies:S-SCDT-10_1038-S44319-025-00633-z.

## Disclosure and competing interests statement

The authors declare no competing interests.

# Expanded View Figures

**Figure EV1. Assessing the germline competence of Stella-eGFP ESCs and confirming NANOG's regulation of *Prdm1* and *Prdm14*.**

(A) Chimeras generated by Stella-eGFP ESCs. (B) Representative of germline transmission of chimeras from Stella-eGFP ESCs. (C) Summary of chimeric mice generated by blastocyst microinjection. (D) Genotyping of pups to detect transgene originating from Stella-eGFP ESCs. (E) Scheme for CRISPR/Cas9-assisted transgene integration into the H11 "safe harbor" locus. (F) Overview of how ESCs were generated that contain NANOG-responsive reporters (*Prdm1* and *Prdm14* enhancers upstream of a *Shh* basal promoter) at the H11 locus. (G) PCR verification of clones harboring knock-in vectors. (H) Representative β-gal staining of EBs. (I) Schema illustrating key TFs acting during PGC development. Scale bar in (H) represents 50 μm. (J) Reverse transcription PCR of mRNAs in day 2 EBs.

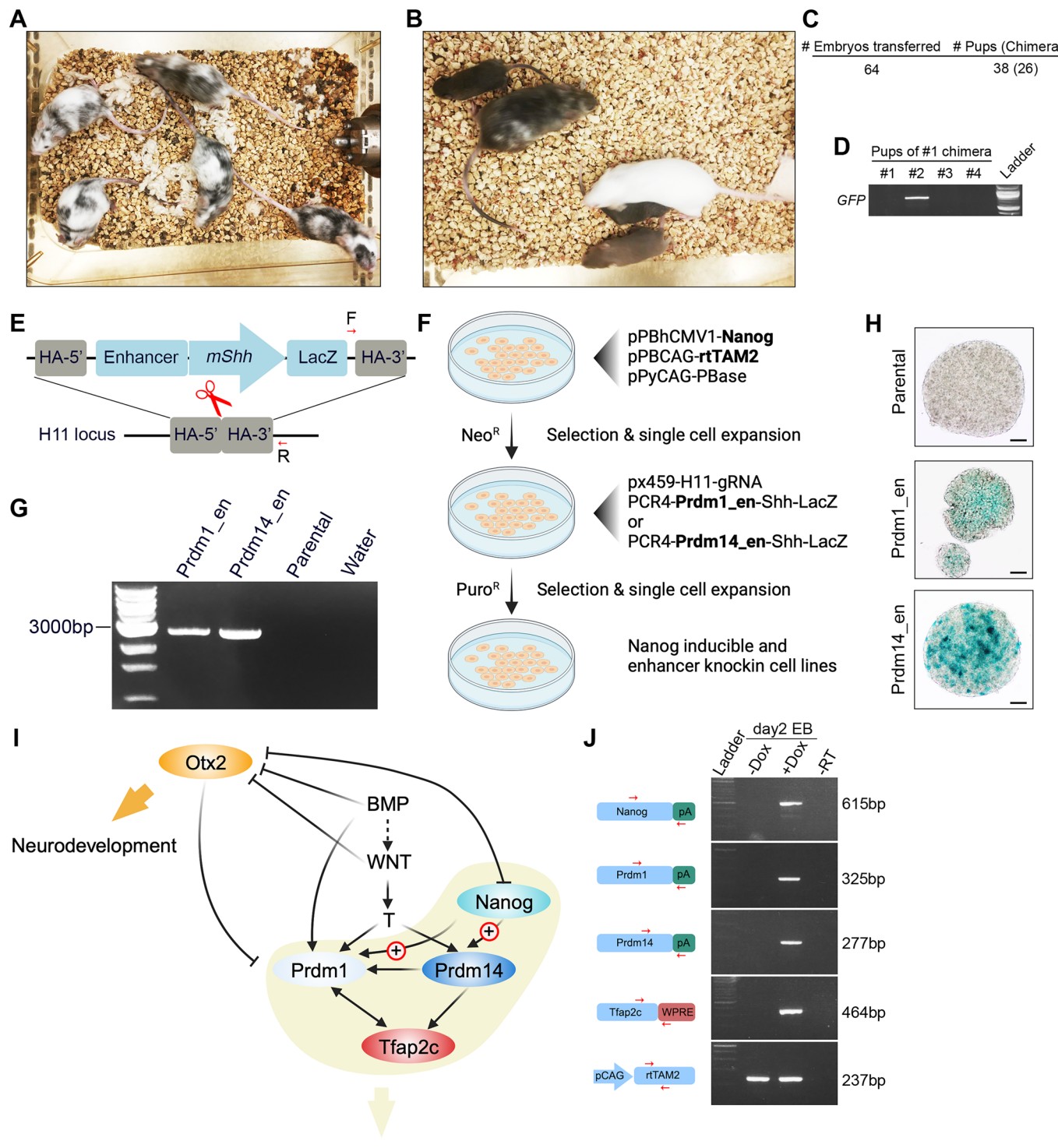

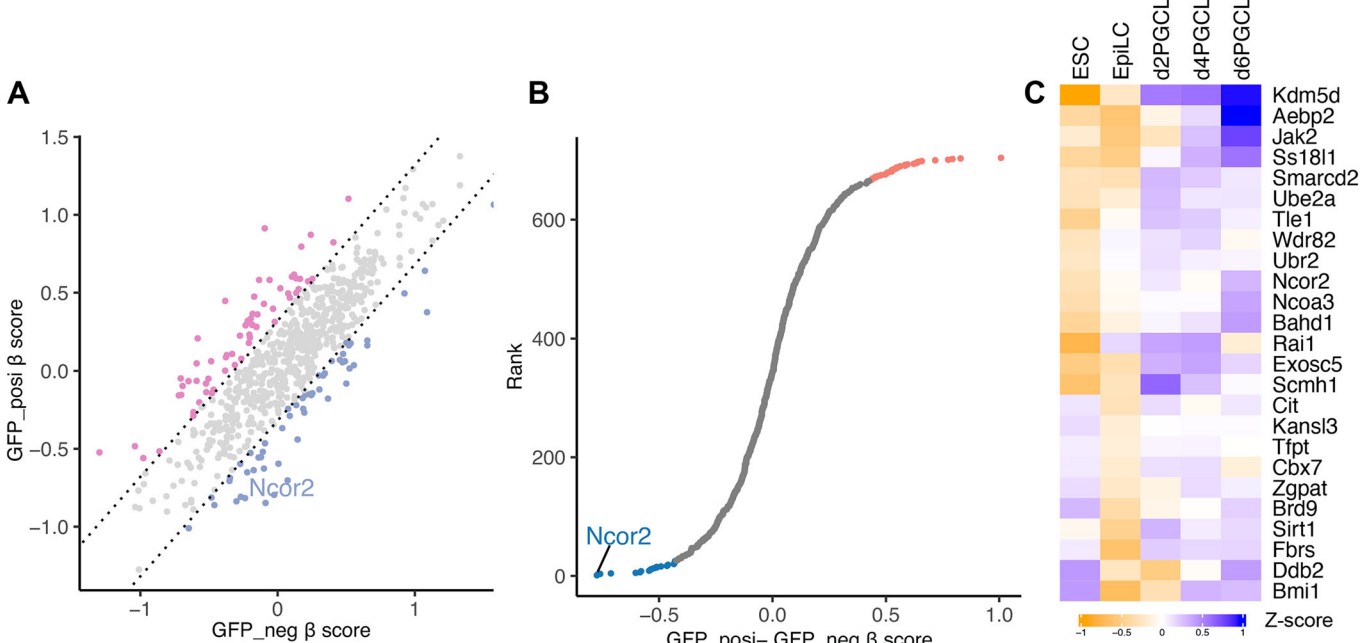

**Figure EV2.  CRISPRi screen epigenetic genes affecting PGCLC differentiation.**

(A) Scatterplot of Stella-eGFP⁺ and Stella-eGFP⁻ β scores of each gene. The β scores were normalized using non-targeting sgRNA sequences. The two dashed lines indicates ± 1 S.D. of the differences between the Stella-eGFP⁺ and Stella-eGFP⁻ cells β scores. (B) Rank plot of the differential β scores, calculated by subtracting Stella-eGFP⁻ β scores from Stella-eGFP⁺ β scores. The color scheme of the dots is the same as in (A). (C) Gene expression heatmap of 25 candidate genes with β scores and exhibited increased expression during PGCLC differentiation compared to ESC and EpiLC stages.

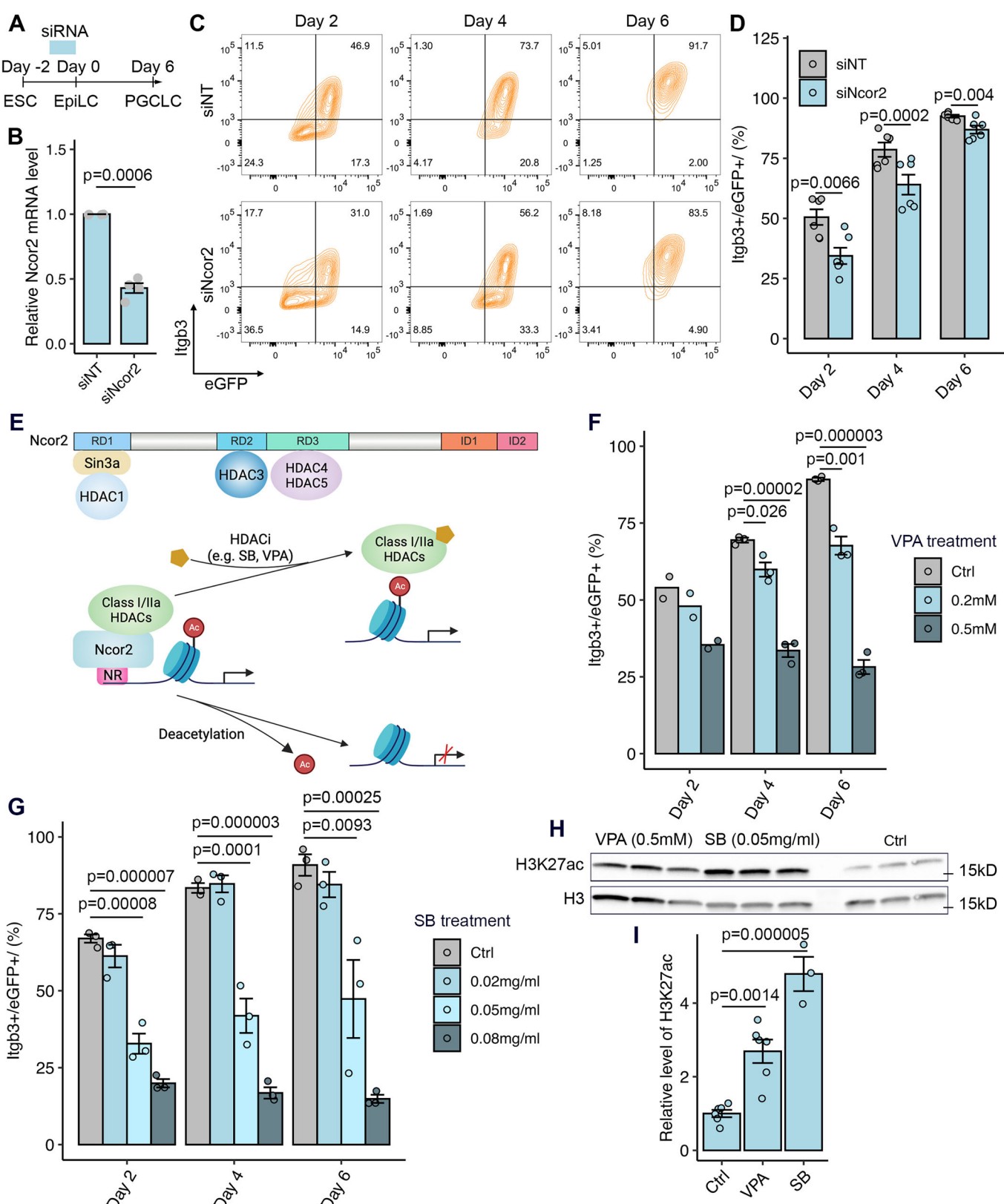

**Figure EV3.  NCOR2 deficiency and HDAC inhibitors suppress PGCLC differentiation in vitro.**

(A) Experimental schema showing siRNA treatment. (B) Effectiveness of *Ncor2* siRNA on day 2 after PGCLC induction. $n = 3$ biological replicates. mRNA levels quantified by qRT-PCR. siNT, non-targeting siRNA. (C) Relative population density in EBs treated with siNT or siNcor2. siNcor2, siRNA targeting *Ncor2*. (D) Percentages of Itgb3$^+$/eGFP$^+$ cells in knockdown (siNcor2) vs control (siNT) groups at the indicated stages after transfection of small RNAs. $n = 6$ biological replicates. eGFP, Stella-eGFP. (E) Structure of NCOR2 and interaction with HDACs (upper panel). Impact of HDAC inhibitors on NCOR2-HDAC complexes (lower panel). (F) Percentage of Itgb3$^+$/eGFP$^+$ populations within EBs treated with water (Ctrl) or VPA as assessed by flow cytometry. $n = 3$ biological replicates except there are 2 biological replicates for day 2. (G) Same as (F), except for SB treatment. $n = 3$ biological replicates. (H) Western blot analysis of H3K27ac and H3 protein levels in day 2 Stella-eGFP$^+$ cells treated with HDACi. (I) Quantification of relative H3K27ac protein levels normalized to H3. $n \geq 3$ biological replicates. Data in (B, D, F, G, I) are represented as the mean ± SEM. Data in (B, D) were analyzed using a two-tailed paired *t* test, and data in (F, G, I) were analyzed using one-way ANOVA with Tukey's *post hoc* test.

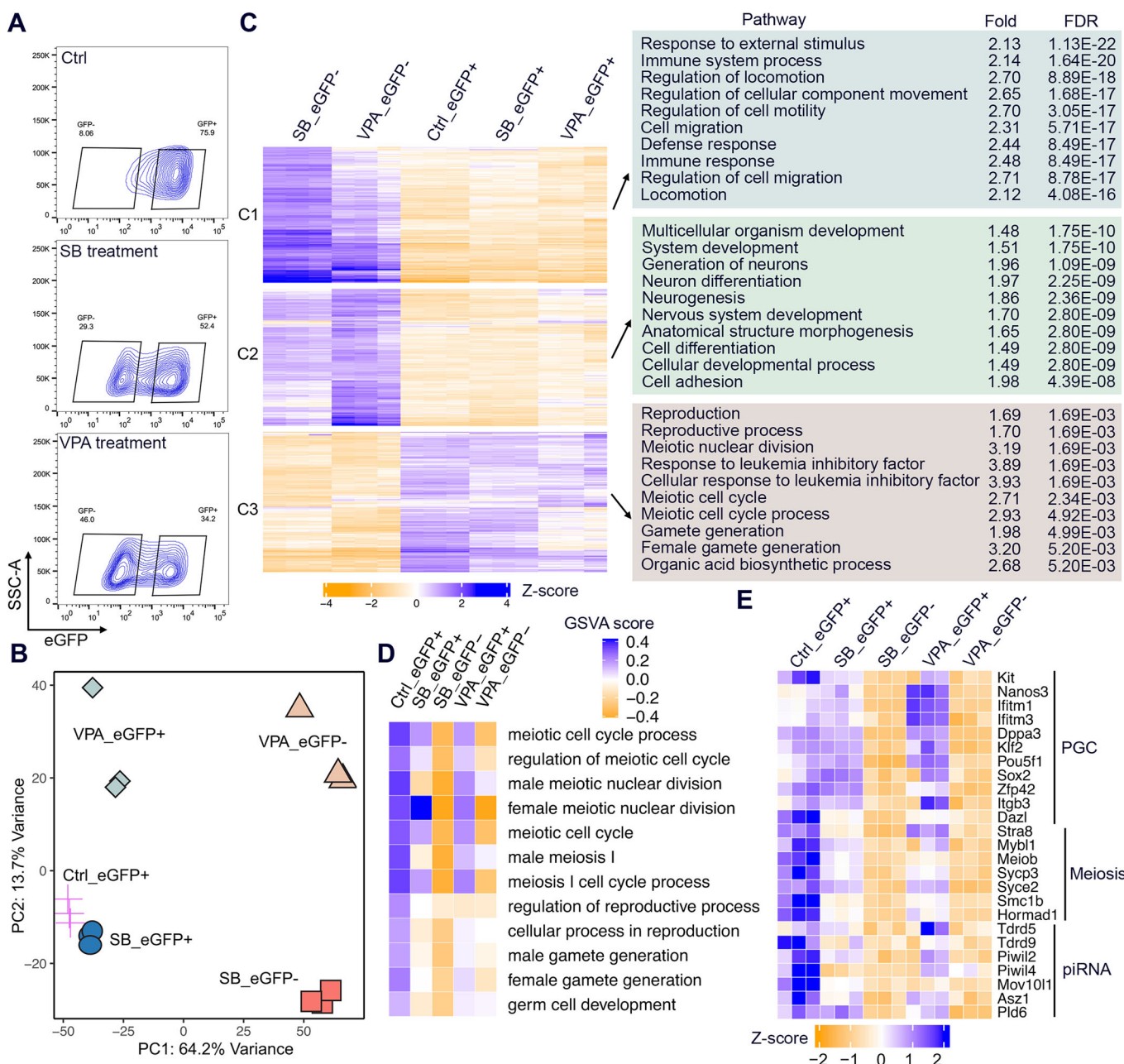

**Figure EV4. Downregulation of germline networks by HDAC inhibitors.**

(A) Flow sorting gates for Stella-eGFP⁺ and Stella-eGFP⁻ cells from day 6 EBs used for RNA-seq. (B) PCA of transcriptomes of sorted cells of indicated groups. SB, sodium butyrate; VPA, valproic acid; Ctrl, control untreated. (C) Heatmap of k-Means clustering of variably expressed genes in SB, VPA and Ctrl cells ($n = 2,000$; $k = 3$). Genes were grouped into 3 clusters ("C1-3") based on expression similarity. Top enriched GO terms for the genes in each cluster shown with fold enrichment (Fold) and false discovery rate (FDR). (D) Heatmap showing the average GSVA enrichment score of selected germline development pathways. (E) Heatmaps of normalized RNA-seq reads for selected germline genes.

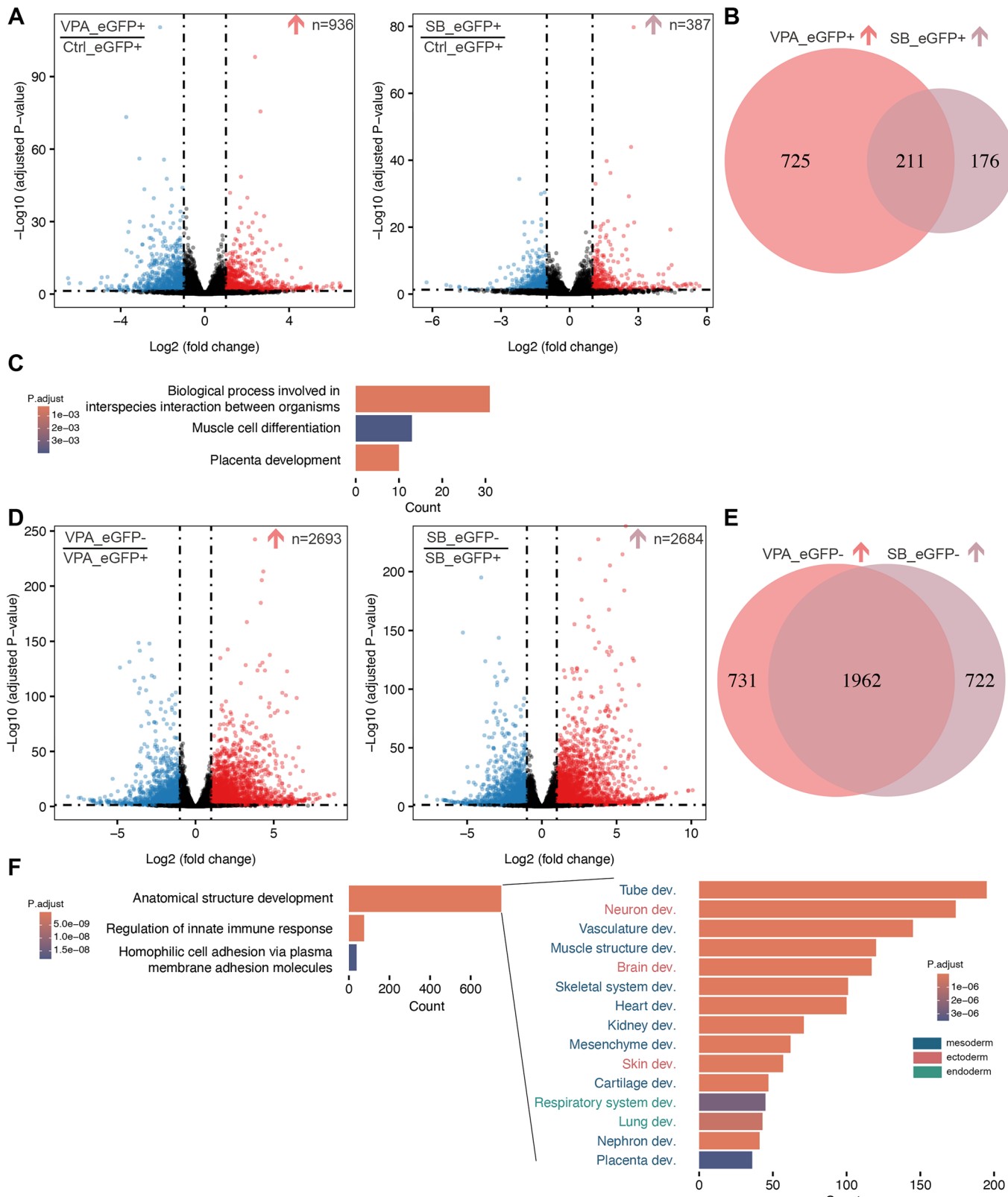

◀   **Figure EV5.  Evidence of somatic lineage differentiation induced by HDACi treatment of differentiating PGCLC cultures.**

(A) Volcano plots of $\log_2$ (fold change) versus $-\log_{10}$ (adjusted $P$ value) for VPA_eGFP[+] (left) or SB_eGFP[+] (right) versus Ctrl_ eGFP[+] cells. (B) Venn diagrams of the upregulated DEGs from (A). (C) Top enriched GO terms for the commonly upregulated genes in eGFP[+] cells in HDACi treatment groups. (D) Volcano plots of Stella-eGFP[-] versus Stella-eGFP[+] cells treated with VPA (left) or SB (right). (E) Venn diagram of the commonly upregulated DEGs from (D). (F) Left: Top enriched GO terms for the HDACi upregulated genes from (E). Right: Count distribution of the "child" GO terms under "Anatomical structure development".

