## [Peer Review File · EMBO Reports]

A primordial germ cell-like-cell platform enables CRISPRi screen for epigenetic fertility modifiers

Liangdao Li, Jingyi Gao, Dain Yi, Alex Sheft, John Schimenti, and Xinbao Ding

Corresponding author(s): John Schimenti (jcs92@cornell.edu)

Review Timeline:

Transfer Date:	2nd Jul 25
Editorial Decision:	8th Jul 25
Revision Received:	4th Sep 25
Editorial Decision:	10th Oct 25
Revision Received:	20th Oct 25
Accepted:	28th Oct 25

Editor: Achim Breiling

Transaction Report: This manuscript was transferred to EMBO reports following peer review at Review Commons.

**Review
COMMONS**

Dear Prof. Schimenti,

Thank you for transferring your manuscript to from Review Commons to EMBO reports. I now went through the manuscript, the referee reports from Review Commons (attached again below) and your revision plan.

Judging from your revision plan, it seems that you will be able to adequately address the concerns of the referees during a major revision. I thus invite you to revise your manuscript accordingly with the understanding that all referee concerns must be addressed in the revised manuscript and/or in a final detailed point-by-point response, as indicated in your revision plan.

Revised manuscripts should be submitted within three months of a request for revision. Please contact me to discuss the revision (also by video chat) if you have further questions or comments regarding the revision, or should you need additional time.

Acceptance of your manuscript will depend on a positive outcome of another round of review at EMBO reports, using the same referees.

1) a .docx formatted version of the final manuscript text (including legends for main figures, EV figures and tables), but without the figures included. Please make sure that changes are highlighted to be clearly visible. Figure legends should be compiled at the end of the manuscript text.

2) individual production quality figure files as .eps, .tif, .jpg (one file per figure), of main figures and EV figures. Please upload these as separate, individual files upon re-submission. Please make sure that all figure panels are called out separately and sequentially in the manuscript text

For more details please refer to our guide to authors:

See also our guide for figure preparation:

Moreover, please consult our guidelines for figure legend preparation:

4) a complete author checklist, which you can download from our author guidelines (<https://www.embopress.org/page/journal/14693178/authorguide>). Please insert page numbers in the checklist to indicate where the requested information can be found in the manuscript. The completed author checklist will also be part of the RPF.

5) that primary datasets produced in this study (e.g. RNA-seq, ChIP-seq and array data) are deposited in an appropriate public

database. This is now mandatory (like the COI statement). If no primary datasets have been deposited in any database, please state this in this section (e.g. 'No primary datasets have been generated and deposited').

The accession numbers and database should be listed in a formal "Data Availability " section (placed after Materials & Methods) that follows the model below. Please note that the Data Availability Section is restricted to new primary data that are part of this study.

Data availability

6) We now request the publication of original source data with the aim of making primary data more accessible and transparent to the reader. You will receive a separate email with instructions for providing source data with your revised manuscript, including information how to upload and organize the files.

8) Regarding data quantification and statistics, please make sure that the number "n" for how many independent experiments were performed, their nature (biological versus technical replicates), the bars and error bars (e.g. SEM, SD) and the test used to calculate p-values is indicated in the respective figure legends (also for EV and Appendix figures). Please also check that all the p-values are explained in the legend, and that these fit to those shown in the figure. Please provide statistical testing where applicable. Please avoid the phrase 'independent experiment', but clearly state if these were biological or technical replicates. Please also indicate (e.g. with n.s.) if testing was performed, but the differences are not significant. In case n=2, please show the data as separate datapoints without error bars and statistics. See also: <http://www.embopress.org/page/journal/14693178/authorguide#statisticalanalysis>

9) Please add scale bars of similar style and thickness to microscopic images, using clearly visible black or white bars (depending on the background). Please place these in the lower right corner of the images themselves. Please do not write on or near the bars in the image but define the size in the respective figure legend.

10) Please also note our reference format:

12) We now use CRediT to specify the contributions of each author in the journal submission system. CRediT replaces the author contribution section. Please use the free text box to provide more detailed descriptions and do NOT provide your final manuscript text file with an author contributions section. See also our guide to authors: <https://www.embopress.org/page/journal/14693178/authorguide#authorshipguidelines>

13) All Materials and Methods need to be described in the main text using our 'Structured Methods' format, which is required for all research articles. According to this format, the Methods section should include a Reagents and Tools Table (listing key

reagents, experimental models, software, and relevant equipment and including their sources and relevant identifiers), uploaded as separate file, and a Methods section in which we encourage the authors to describe their methods using a step-by-step protocol format with bullet points, to facilitate the adoption of the methodologies across labs. More information on how to adhere to this format as well as downloadable templates (.doc) for the Reagents and Tools Table can be found in our author guidelines (section 'Structured Methods'):

14) Please add up to 5 keywords to the manuscript and order the sections like this, using these names:

Title page - Abstract (not more than 175 words) - Keywords - Introduction - Results - Discussion - Methods - Data availability section - Acknowledgements (please include here also the funding information) - Disclosure and Competing Interests Statement - References - Figure legends - Expanded View Figure legends

15) Please make sure that all the funding information is also entered into the online submission system and that it is complete and similar to the one in the acknowledgement section of the manuscript text file.

I look forward to seeing a revised version of your manuscript when it is ready. Please let me know if you have questions or comments regarding the revision.

Kind regards,

Achim

Referee #1:

This manuscript presents a scalable, cytokine-free platform for generating mouse primordial germ cell-like cells (PGCLCs) from either epiblast-like cells (EpiLCs) or formative embryonic stem cells (ESCs) via doxycycline-inducible overexpression of four transcription factors: Nanog, Prdm1, Prdm14, and Tfap2c (the 4TF system). The authors demonstrate high-efficiency PGCLC induction (~80% Stella-eGFP+ cells) and show that the resulting cells exhibit transcriptomic and epigenetic features resembling late-stage in vivo PGCs. These PGCLCs can further differentiate into spermatogonia-like cells in vitro when aggregated with neonatal testicular somatic cells, although the extent and robustness of this differentiation remain to be fully validated. Using this platform, the authors perform a high-throughput CRISPRi screen to identify epigenetic regulators of PGCLC formation, notably highlighting Ncor2 as a critical factor. They further report that in utero exposure to histone deacetylase (HDAC) inhibitors, such as valproic acid and sodium butyrate, impairs PGCLC induction and results in hypospermatogenesis in adult males, though the mechanistic link to PGC loss in vivo requires stronger evidence. Overall, this study introduces an innovative, cost-effective, and scalable method for PGCLC induction, combines transcriptomic, functional, and in vivo analyses, and offers translational relevance by connecting epigenetic regulation to reproductive outcomes. The 4TF platform also enables large-scale functional genomic screens in germ cell biology, addressing longstanding technical barriers related to efficiency and cell numbers.

****Major comments:****

1. Lines 35-36: The role of Nanog is described, but the functions of the other three transcription factors (Prdm1, Prdm14, and Tfap2c) are not discussed here. It would be helpful to briefly mention their contributions to PGCLC induction.
2. Line 38: The phrase "more advanced than cytokine-induced PGCLCs" is ambiguous. Do the authors mean that these PGCLCs are more developmentally progressed or that they more closely resemble in vivo PGCs? Please clarify.

3. Line 38: The term "these differentiated PGCLCs" may be misleading-should this be "induced PGCLCs" instead?
4. Line 40: "epigenome-wide CRISPRi"? Should be "CRISPRi for epigenetic factors".
5. Line 46: Consider changing "PGCLC differentiation" to "PGCLC formation" for accuracy.
6. The statement "Exposure of developing mouse embryos to SB or VPA caused hypospermatogenesis" (Lines 45-46) seems like an overinterpretation. The link between HDAC inhibition during fetal development and impaired PGC formation is suggested, but not directly demonstrated. Alternative interpretations, such as postnatal effects on spermatogonial stem cells, should be acknowledged.
7. Line 194: It is essential to include Western blot or other protein-level validation to confirm successful Dox-induced overexpression of the four transcription factors (Nanog, Prdm1, Prdm14, and Tfap2c). This is critical for interpreting the efficiency and specificity of the 4TF induction system.
8. Line 200: The data showing higher H3K27me3 in GFP-dim cells suggest that epigenetic reprogramming is initiated early. However, GFP-bright cells do not appear to show a further increase in H3K27me3 compared to GFP-dim cells, raising questions about the correlation between GFP intensity and progressive epigenetic maturation. Additional quantification or clarification would strengthen this conclusion.
9. Line 304: The phrase "certain DDX4-positive cells" is vague. Please provide quantitative data showing the percentage of DDX4+ cells that also express PLZF. Since PLZF+ cells appear to be rare in the presented image, what is the identity of the DDX4+/GCNA+ cells that do not express PLZF? Additionally, it would be informative to quantify PLZF+ cells at different time points indicated in Figure 4A-specifically, before aggregation with testicular somatic cells and after two weeks of co-culture.
10. Line 310: Have the authors tested whether retinoic acid treatment of induced PGCLCs upregulates Stra8 expression? In addition, both teratoma cells/ES cells can express Stra8. Thus, the use of Stra8 does not support the conclusion regarding further developmental potential.
11. Line 311: The statement that "4TF-induced PGCLCs have the capacity of further development in vitro" would be strengthened by transcriptomic analysis of the post-aggregation PGCLCs. This would help determine whether their gene expression profile aligns with spermatogonial or pre-meiotic stages.
12. Lines 397-412: To more directly assess whether fetal exposure to HDAC inhibitors impairs PGC formation, gonads should be collected from E12.5 embryos for analysis. This would allow direct quantification of PGC numbers and identity during the developmental window when such effects are likely to manifest.
13. Lines 410-412: There are no experimental evidence to support these claims "These findings suggest that fetal exposure to SB and VPA leads to PGC loss, or possibly spermatogonial stem cell loss, contributing to hypospermatogenesis in adulthood".
14. As pharmacological treatment, SB or VPA could operate through other pathways.
15. Line 444-447: How about PLZF expression?

****Minor comments:****

The term "PGCLC differentiation" is used throughout the manuscript, but this is somewhat misleading. Since the process being described refers to the induction of PGCLCs from EpiLCs or formative ESCs, "PGCLC induction" would be more accurate and less ambiguous. Please revise the terminology consistently throughout the manuscript.

****Significance:****

This study presents a well-executed and technically innovative approach to generating PGCLCs at scale with 4 TFs. The integration of a CRISPRi screen to identify epigenetic regulators further adds applicability. However, some conclusions-particularly regarding in vitro differentiation into spermatogonia-like cells and in vivo effects of HDAC inhibitors-require stronger evidence.

Compared to earlier PGCLC induction systems, this 4TF method offers significantly improved efficiency, scalability, and independence from exogenous signaling pathways.

This work will be of broad interest to developmental biologists, reproductive biologists, stem cell researchers, and those in the fields of epigenetics and environmental reproductive toxicology.

Referee #2:

The authors developed a new method to make mouse germ cell-like cells (PGCLCs) by turning on four key genes (Nanog, Prdm1, Prdm14, Tfap2c) in stem cells, without using cytokines. They use a Stella-eGFP reporter to track germ cell identity and confirm PGCLC formation through flow cytometry, gene expression, and epigenetic changes. The system is used to run a CRISPRi screen targeting ~700 epigenetic regulators, identifying NCOR2 as important for germ cell development. Blocking NCOR2 or HDAC activity reduces PGCLC formation. Treating mice with HDAC inhibitors during pregnancy also leads to reduced sperm production in the offspring.

****Major comments:****

1. Some claims need clarification or evidence. For example, the statement that 4TF PGCLCs are "more advanced" than cytokine PGCLC (line 38) needs to be defined (e.g. expression of specific late-PGC markers). The assertion that the platform is "cost-effective" is not substantiated by data or analysis. Also, it is not shown whether the platform can be scaled up beyond 6-well plates or miniaturized for high-throughput screening. Any claim about scalability or throughput should be backed by statistical details, including experimental sample sizes, litter sizes etc.

2. Forcing high levels of Nanog plus three PGC factors may bypass normal developmental cues. It is possible that some aspects of germ cell specification are artificially induced or that off-target effects occur. For example, Nanog overexpression can broadly suppress differentiation and affect X-inactivation or imprinting (Williams et al., 2011*). The manuscript should address whether the 4TF PGCLCs truly follow a PGC regulatory program or simply reflect a generic primed state. For instance, are key PGC-specific genes (e.g. Dppa3/Stella, Nanos3, Mvh/Ddx4) correctly regulated in their temporal order? The dual IF staining in Figure 4 (e.g. DDX4 with GCNA, PLZF, SOX9, and STRA8) nicely supports germline progression. If triple co-expression of PGC markers such as Stella, Nanos3, and Mvh has also been assessed, highlighting that in the main text would further strengthen the characterization. If not, the authors might consider this as a future enhancement to confirm coordinated marker expression at the single-cell level.

*Williams, L.H., Kalantry, S., Starmer, J., and Magnuson, T. (2011). Transcription precedes loss of Xist coating and depletion of H3K27me3 during X-chromosome reprogramming in the mouse inner cell mass. *Development* 138, 2049-2057.

3. The authors state that 4TF PGCLCs resemble *in vivo* PGCs, but details of this analysis are limited. It is unclear how homogeneous the PGCLC population is or whether subpopulations (e.g. partially differentiated or somatic-like cells) are present. The RNA-seq data should be examined for any residual somatic programs or incomplete reprogramming. Some methodological details could undermine interpretations. For instance, the Stella-eGFP reporter marks a germ cell gene, but Stella alone is not PGC-specific at all stages.

4. The comparisons to "cytokine-induced" PGCLCs need clarification: are these data side-by-side from identical ESC lines and conditions? The baseline efficiency and quality of the cytokine method in the authors' hands is not shown, making it hard to judge the improvement.

5. The finding that maternal HDACi exposure causes hypospermatogenesis is intriguing, but the evidence is tenuous. It is not shown whether this effect is direct on germ cells or secondary to other developmental defects (VPA is a known teratogen). HDAC activity is broadly required for cell differentiation; VPA and sodium butyrate are non-specific and can affect many lineages. The manuscript should distinguish general toxicity or differentiation block from germline-specific effects. Also, NCOR2 itself has many targets - is its effect in PGCLCs mediated only via HDAC recruitment, or through other co-repressors? The conclusion that NCOR2 is an "epigenetic fertility modifier" is plausible but not fully supported without deeper mechanistic insight.

It may help to use more selective HDAC inhibitors or genetic knockdown to pinpoint which HDAC isoforms are critical. For instance, test class I vs II-specific inhibitors, and knockdown Hdac3 (a class I) or Hdac4/5 (class IIa) separately in PGCLCs. Include cell viability and cell cycle assays to ensure that effects are not due to cytotoxicity. Conversely, test whether adding HDAC activators or overexpressing HDAC3/4 can enhance PGCLC formation. Additionally, include a non-germline differentiation control (e.g. differentiate ESCs to neurons or muscle under similar HDACi treatment) to show that the observed effect is germline-specific.

6. The screen identified dozens of "hits" but only NCOR2 is discussed in detail. It's unclear how reproducible the screen was (technical replicates, library coverage, false positives). Were any other top hits followed up or validated individually? If NCOR2 is highlighted, why not other strong hits?

7. The authors could measure expression of somatic genes (e.g. Hoxb1, Hand1) or PGC markers in NCOR2-depleted cells, paralleling Mochizuki's findings. This is partly addressed in RNA-seq of Stella⁺ vs Stella⁻ cells under HDACi (Figs S15-S16). It may suffice to simply clarify in the main text that NCOR2 likely enforces germline fate by repressing somatic programs, as the siRNA and HDACi data imply.

8. To clarify the role of NCOR2 in PGCLC formation, it will be good to independently validate NCOR2's role by CRISPR knockout or RNAi in EpiLCs (rather than CRISPRi), and rescue the phenotype by overexpressing an HDAC-binding-deficient NCOR2 mutant to see if HDAC recruitment is necessary. Additionally, RNA-seq of NCOR2-knockdown PGCLCs could identify which germline and somatic genes are misregulated, helping distinguish primary regulatory effects (see pt#8).

Optional: If the authors wish to further explore chromatin-level mechanisms, ChIP-qPCR for histone acetylation (e.g. H3K27ac) at germline gene loci could help define how NCOR2-HDAC repression shapes gene accessibility. Additionally, while not essential for the core conclusion, assessing whether other top screen hits (~53 total) similarly influence germline-specific versus general differentiation pathways could provide a broader picture of epigenetic regulation in PGCLCs.

9. While the 4TF PGCLCs display germ cell markers and epigenetic hallmarks of PGCs, it is unclear if they can complete germ cell development *in vivo* or in a more physiologically relevant assay. Claims of "spermatogonia-like" differentiation are based on

in vitro morphology or marker expression, but the work lacks definitive evidence (e.g. progression through meiosis, haploid formation, or transplantation assays). Without demonstrating formation of mature germ cells or fertility restoration, the true developmental potential of these PGCLCs remains uncertain.

****Minor comments:****

1. Line 205: The Results mention that "PGCLCs could be dedifferentiated to EGCs" when transferred to 2iLIF (Fig. S2). This is interesting and perhaps should be briefly discussed (even if just in Supp.), as it shows germline epigenetic resetting.
2. The alphanumeric figure labels are not intuitive and may be confusing; for example, in Fig. 2, 'D' appears after 'A' and is followed by 'B' and then 'C'.
3. The term "cost-effective" is used in the Abstract; authors could briefly quantify or justify this (e.g. "using small-molecule versus expensive cytokines").
4. The number of animals, dosage, and controls (e.g. unaffected maternal health) should be detailed.

****Significance:****

This study introduces a simplified and high-efficiency method to generate PGCLCs without cytokines, and combines it with a functional CRISPRi screen to identify chromatin regulators like NCOR2 that affect germ cell fate. The strongest aspects are the robust PGCLC induction and integration of genomic screening with epigenetic relevance. However, some conclusions are premature without stronger mechanistic data, additional validation of screen hits, and functional demonstration of germline potential in vivo.

Audience

Researchers studying germline development, fertility disorders reproductive biology, epigenetics, and stem cell differentiation

Reviewer expertise

Germline development, stem cell biology

Reviewer #1

This manuscript presents a scalable, cytokine-free platform for generating mouse primordial germ cell-like cells (PGCLCs) from either epiblast-like cells (EpiLCs) or formative embryonic stem cells (ESCs) via doxycycline-inducible overexpression of four transcription factors: Nanog, Prdm1, Prdm14, and Tfap2c (the 4TF system). The authors demonstrate high-efficiency PGCLC induction (~80% Stella-eGFP+ cells) and show that the resulting cells exhibit transcriptomic and epigenetic features resembling late-stage *in vivo* PGCs. These PGCLCs can further differentiate into spermatogonia-like cells *in vitro* when aggregated with neonatal testicular somatic cells, although the extent and robustness of this differentiation remain to be fully validated. Using this platform, the authors perform a high-throughput CRISPRi screen to identify epigenetic regulators of PGCLC formation, notably highlighting Ncor2 as a critical factor. They further report that *in utero* exposure to histone deacetylase (HDAC) inhibitors, such as valproic acid and sodium butyrate, impairs PGCLC induction and results in hypospermatogenesis in adult males, though the mechanistic link to PGC loss *in vivo* requires stronger evidence. Overall, this study introduces an innovative, cost-effective, and scalable method for PGCLC induction, combines transcriptomic, functional, and *in vivo* analyses, and offers translational relevance by connecting epigenetic regulation to reproductive outcomes. The 4TF platform also enables large-scale functional genomic screens in germ cell biology, addressing longstanding technical barriers related to efficiency and cell numbers.

Major comments:

1. Lines 35-36: The role of Nanog is described, but the functions of the other three transcription factors (Prdm1, Prdm14, and Tfap2c) are not discussed here. It would be helpful to briefly mention their contributions to PGCLC induction.

Response: Prdm1, Prdm14, and Tfap2c are well-known in the field, and so for the purpose of simplicity in the Abstract, we prefer to leave it as is; rather we have elaborated on their functions in the Introduction (pg. 3, lines 13-16 in updated version), and added some references. Also, we had to substantially downsize the abstract to comply with journal guidelines.

2. Line 38: The phrase "more advanced than cytokine-induced PGCLCs" is ambiguous. Do the authors mean that these PGCLCs are more developmentally progressed or that they more closely resemble *in vivo* PGCs? Please clarify.

Response: Sorry for the lack of clarity. The answer is essentially both. They are more advanced developmentally in terms of transcriptomics when compared to transcriptomes of various stage PGCs (see Fig. 2). We revised this phrase to "more advanced developmental stage than cytokine-induced PGCLCs".

3. Line 38: The term "these differentiated PGCLCs" may be misleading-should this be "induced PGCLCs" instead?

Response: Done.

4. Line 40: "epigenome-wide CRISPRi"? Should be "CRISPRi for epigenetic factors".

Response: We changed to, "... to conduct a CRISPRi screen of 701 known epigenetic genes to identify...."

5. Line 46: Consider changing "PGCLC differentiation" to "PGCLC formation" for accuracy.

Response: Done.

6. The statement "Exposure of developing mouse embryos to SB or VPA caused hypospermatogenesis" (Lines 45-46) seems like an overinterpretation. The link between HDAC inhibition during fetal development and impaired PGC formation is suggested, but not directly demonstrated. Alternative interpretations, such as postnatal effects on spermatogonial stem cells, should be acknowledged.

Response: We appreciate the reviewer's comment. We revised the Abstract to read "... valproic acid and sodium butyrate suppressed PGCLC formation and sperm counts of *in utero*-exposed animals.". Additionally, we revised the Discussion (pg. 21, lines 18-21 in updated version) to read, "However, further studies are needed to elucidate the mechanisms (specific roles of HDAC isoforms and other NCOR2 co-repressors), developmental stages, and cell types (e.g., PGCs, SSCs) that are impacted by HDAC inhibitors to yield the observed phenotypes."

7. Line 194: It is essential to include Western blot or other protein-level validation to confirm successful Dox-induced overexpression of the four transcription factors (Nanog, Prdm1, Prdm14, and Tfap2c). This is critical for interpreting the efficiency and specificity of the 4TF induction system.

Response: PGCLC induction was dependent upon Dox addition (Fig. 1D and E; 5D and E; 7C and E; S9), and both independently-derived 4TF cell lines exhibited consistently higher induction efficiencies than cell lines expressing fewer than 4 TFs (Fig. 1A). Nevertheless, we appreciate the reviewer's comment that it is conceivable that not all TFs were induced. To address this, we performed RT-PCR of the TFs (using transgene-specific primers), and confirmed robust RNA expression following Dox addition (see Fig. EV1J). Although the reviewer suggested Western blotting to quantify transgenic TF proteins, this would not distinguish between endogenous and transgenic proteins.

8. Line 200: The data showing higher H3K27me3 in GFP-dim cells suggest that epigenetic reprogramming is initiated early. However, GFP-bright cells do not appear to show a further increase in H3K27me3 compared to GFP-dim cells, raising questions about the correlation between GFP intensity and progressive epigenetic maturation. Additional quantification or clarification would strengthen this conclusion.

Response: We appreciate the reviewer's important point. Our data indeed indicate a correlation between GFP intensity and epigenetic maturation. One possibility is that the dim population actually consists of cells early in the progression to PGCLCs as well as later, although we can't tell from this experiment. We note (although it wasn't originally clear) that these cells were scored by dispersing EBs into single cells, attached them to a slide, and scanning fluorescence by Image J. Because the classification of dim vs bright is somewhat arbitrary, in the revision we will simply group them as GFP- vs GFP+ and re-did the panel (Fig. 1F) and the stats, and edited the text (p.g. 8, lines 13-14) accordingly.

9. Line 304: The phrase "certain DDX4-positive cells" is vague. Please provide quantitative data showing the percentage of DDX4+ cells that also express PLZF. Since PLZF+ cells appear to be rare in the presented image, what is the identity of the DDX4+/GCNA+ cells that do not express PLZF? Additionally, it would be informative to quantify PLZF+ cells at different time points indicated in Figure 4A-specifically, before aggregation with testicular somatic cells and after two weeks of co-culture.

Response: Sorry for the confusion. We quantified the proportion of DDX4+ cells co-expressing PLZF at two time points. At 1wk of co-culture, no DDX4+/PLZF+ double-positive cells were detected. At 2wk of co-culture, 8.4% of DDX4+ cells also expressed PLZF. These results indicate that a subset of DDX4+ cells begin to acquire PLZF expression after extended interaction with the somatic niche. We have incorporated these quantitative findings in the revised manuscript (p.g. 11, line 30-31 in revised version). The DDX4+/GCNA+/PLZF- population likely represents germ cells that have progressed beyond the PGC stage and acquired early germline identity, but have not yet fully transitioned to the spermatogonial state marked by PLZF. We added discussion on p.g. 20, lines 5-7 in the revision.

10. Line 310: Have the authors tested whether retinoic acid treatment of induced PGCLCs upregulates Stra8 expression? In addition, both teratoma cells/ES cells can express Stra8. Thus, the use of Stra8 does not support the conclusion regarding further developmental potential.

Response: We thank the reviewer for this important and nuanced comment regarding the interpretation of Stra8 expression as an indicator of germ cell developmental progression. In our study, we adapted an established testis organ culture system in which the culture medium lacks exogenous RA (1). Under these conditions, the co-cultured testicular somatic cells are likely to produce sufficient endogenous RA to trigger meiotic entry in germ cells. The culture condition was described in the section "Cell aggregate cultures for further differentiation" in M&M. To demonstrate additional developmental potential, we use GCNA+/STRA8+ cells to indicate the presence of meiotic germ cells within the co-cultured aggregated (Fig. 4D and 7E). GCNA is a well-established marker of germ cells from E11 onward (2,3), providing additional support for the germ cell identity and developmental progression in our system. Nevertheless, on lines 7-9 of p.g. 12 we state that presence of GCNA+/STRA8+ cells "suggest.... capacity for further development in vitro" and on line 11 of p.g. 20 state that "further work is necessary to explore whether these are truly meiotic cells ...". We hope this conveys sufficient caution.

11. Line 311: The statement that "4TF-induced PGCLCs have the capacity of further development in vitro" would be strengthened by transcriptomic analysis of the post-aggregation PGCLCs. This would help determine whether their gene expression profile aligns with spermatogonial or pre-meiotic stages.

Response: Indeed it would be informative to go beyond the antibody marker analyses with single cell transcriptome analysis to determine cell identities, and we are working on developing a platform to improve differentiation to SSCs and beyond, but this is a problem that will not be solved soon at high efficiency. To hopefully allay the reviewer's concerns, we have tempered the statement to read (p.g. 12, lines 8-10), "These observations suggest that 4TF-induced PGCLCs have some capacity for further development *in vitro*, but optimization of conditions and functional assays will be required to determine their full potential."

12. Lines 397-412: To more directly assess whether fetal exposure to HDAC inhibitors impairs PGC formation, gonads should be collected from E12.5 embryos for analysis. This would allow direct quantification of PGC numbers and identity during the developmental window when such effects are likely to manifest.

Response: Indeed quantification of PGCs in vivo following HDACi could confirm the in vitro results where VPA and SB inhibited PGCLC formation and disrupted the germline transcriptome. However, these are not trivial endeavors (PGC quantification by serial sectioning is tedious), and the student (the first author) who performed these experiments has graduated, so these

experiments cannot be done in a timely manner. Instead, we revised the indicated paragraph in Results to read, "These findings suggest that fetal exposure to SB and VPA leads to decreased sperm counts and increased atrophic seminiferous tubules in adulthood, possibly a consequence of perturbations to PGCs or SSCs." Additionally, we suggested some alternative possibilities for observed phenotypes in the Discussion on p.g. 21, lines 18-24.

13. Lines 410-412: There are no experimental evidence to support these claims "These findings suggest that fetal exposure to SB and VPA leads to PGC loss, or possibly spermatogonial stem cell loss, contributing to hypospermatogenesis in adulthood".

Response: We agree with the need to be more cautious with interpretation. This comment is similar to points #6 and #12, the responses to which added caution relevant to this issue. Additionally, the following statement was added on p.g. 15, lines 20-22: "These findings suggest that fetal exposure to SB and VPA leads to decreased sperm counts and increased atrophic seminiferous tubules in adulthood, possibly a consequence of perturbations to PGCs or SSCs".

14. As pharmacological treatment, SB or VPA could operate through other pathways.

Response: In the Discussion, we added this sentence at the end of a paragraph (p.g. 21, lines 21-24 in updated version) discussing the effects of SB and VPA: "Although the HDACi-treated PGCLC induction cultures indeed showed increased H3K27ac levels (Fig. S14F-I), it remains to be seen if the in vivo exposure caused the spermatogenesis phenotypes strictly via this epigenetic mechanism."

15. Line 444-447: How about PLZF expression?

Response: We performed PLZF labeling of the somatic cell/PGCLC aggregates derived from formative ESCs: see Fig.7E.

Minor comments:

The term "PGCLC differentiation" is used throughout the manuscript, but this is somewhat misleading. Since the process being described refers to the induction of PGCLCs from EpiLCs or formative ESCs, "PGCLC induction" would be more accurate and less ambiguous. Please revise the terminology consistently throughout the manuscript.

Response: We did as suggested.

Reviewer #2

The authors developed a new method to make mouse germ cell-like cells (PGCLCs) by turning on four key genes (Nanog, Prdm1, Prdm14, Tfap2c) in stem cells, without using cytokines. They use a Stella-eGFP reporter to track germ cell identity and confirm PGCLC formation through flow cytometry, gene expression, and epigenetic changes. The system is used to run a CRISPRi screen targeting ~700 epigenetic regulators, identifying NCOR2 as important for germ cell development. Blocking NCOR2 or HDAC activity reduces PGCLC formation. Treating mice with HDAC inhibitors during pregnancy also leads to reduced sperm production in the offspring.

Major comments:

1. Some claims need clarification or evidence. For example, the statement that 4TF PGCLCs are "more advanced" than cytokine PGCLC (line 38) needs to be defined (e.g. expression of specific late-PGC markers). The assertion that the platform is "cost-effective" is not substantiated by data or analysis. Also, it is not shown whether the platform can be scaled up beyond 6-well plates or miniaturized for high-throughput screening. Any claim about scalability or throughput should be backed by statistical details, including experimental sample sizes, litter sizes etc.

Response: Thanks for these comments which allow us to clarify these key points:

- Regarding the 4TF PGCLCs being "more advanced," this was also raised by Reviewer 1 in major point #2 and was addressed above. The 4TF PGCLCs are more advanced developmentally in terms of transcriptomics when compared to transcriptomes of various stage PGCs (see Fig. 2). We revised this phrase to "more advanced developmental stage than cytokine-induced PGCLCs".

- Regarding "cost-effective," we now elaborate. To summarize: Preparing 10ml of PGCLC differentiation media using cytokines can cost up to \$456 (based on small aliquots of BMP4). Considering that the differentiation efficiency is far lower in cytokine-based protocols, generating an equivalent number of PGCLCs using cytokines would be over 500 times more expensive than using the 4TF system. We updated the comparisons of "cost-effectiveness" for each platform in Fig. S16 and the corresponding figure legend.

- Regarding scalability beyond 6 well plates: we did not use 6-well plates for PGCLC differentiation and have double-checked the manuscript without finding mention. Perhaps the reviewer intended to refer to 12-well plates? To demonstrate scalability, we systematically tested the 4TF PGCLC induction protocol across multiple formats, including 100 mM Petri dishes. To summarize:

- a. Plate and dish: PGCLC differentiation was successfully performed in 96-well, 12-well, and 100 mm Petri dishes. Our system is compatible with both small- and large-scale applications (Fig. 5A-C). The scalability of our platform enabled us to perform large-scale CRISPRi screening using the 100mm Petri dishes (p.g. 14, lines 6-7 in updated version). We note that Reviewer #1 commented that this platform was "... addressing longstanding technical barriers related to efficiency and cell numbers".
- b. Efficiencies: The percentage of PGCLCs (ITGB3+/SSEA1+ or ITGB3+/Stella-eGFP+) was consistently high across all tested formats, with optimal differentiation observed in 12-well plates and slightly reduced efficiency in 96-well plates. Perhaps the latter format is what the reviewer is suggesting about miniaturization for potential high throughput (rather than bulk?) screening?
- c. Statistical details: For each format, we performed at least 3 biological replicates (Fig. 5B). The number of replicates for each group were added to the Fig. 5B figure legend.

2. Forcing high levels of Nanog plus three PGC factors may bypass normal developmental cues. It is possible that some aspects of germ cell specification are artificially induced or that off-target effects occur. For example, Nanog overexpression can broadly suppress differentiation and affect X-inactivation or imprinting (Williams et al., 2011*). The manuscript should address whether the 4TF PGCLCs truly follow a PGC regulatory program or simply reflect a generic primed state. For instance, are key PGC-specific genes (e.g. Dppa3/Stella, Nanos3, Mvh/Ddx4) correctly regulated in their temporal order? The dual IF staining in Figure 4 (e.g. DDX4 with

GCNA, PLZF, SOX9, and STRA8) nicely supports germline progression. If triple co-expression of PGC markers such as Stella, Nanos3, and Mvh has also been assessed, highlighting that in the main text would further strengthen the characterization. If not, the authors might consider this as a future enhancement to confirm coordinated marker expression at the single-cell level.

*Williams, L.H., Kalantry, S., Starmer, J., and Magnuson, T. (2011). Transcription precedes loss of Xist coating and depletion of H3K27me3 during X-chromosome reprogramming in the mouse inner cell mass. *Development* 138, 2049-2057.

Response: We appreciate the reviewer's concern and would like to highlight that Nanog plays critical, context-dependent roles in pluripotency maintenance and PGC specification. In ESCs and iPSCs, Nanog cooperates with Oct4 and Sox2 to sustain pluripotency (4), but during PGC differentiation, Nanog uniquely: a. inhibits non-germline lineages (such as neurodevelopmental pathways, Fig. 2G and H; Fig. S3); b. enhances the PGC regulatory network (Fig. 1E-H; (5)); and c. maintains the PGC fate (Fig. 5F-I; (6)). These functions contribute to the high efficiency, advanced developmental stage and long-term culture of PGCLCs generated in our system compared to cytokine-induced PGCLCs.

We appreciate that reviewer noted the Williams et al paper, but the ESC line used in our study is male, so X-chromosome reprogramming is not involved.

We believe that our manuscript clearly addressed that the 4TF PGCLCs truly follow a PGC regulatory program, not a generic primed state, based on following:

- Transcriptomic analyses demonstrate that 4TF-induced PGCLCs closely resemble late-stage (E10.5–E11.5) *in vivo* PGCs with clear separation from naïve and formative cells.
- RNA-seq data confirm robust, temporally ordered upregulation hallmark of PGC genes in 4TF-induced PGCLCs. Early PGC genes (such as Prdm1, Prdm14, Dnd1, Tfap2c, Dppa3, Itgb3, Phox9/6/5, and Gm9) are expressed initially, followed by late PGC (such as, Dazl, Ddx4, Gata3, Asz1, Dmrt1, and Tex11) and meiosis genes (such as, Stra8, Sycp1/2/3, Meiob, and Smc1b) as differentiation progresses (Fig. 3E), mirroring *in vivo* PGC development.
- The 4TF-induced PGCLCs display loss of H3K9me2, gain of H3K27me3, and demethylation at imprinted loci, mirroring authentic PGCs (7).
- Primed genes (such as, Sfrp2, Sox11 and Jakmip2) were not enriched in 4TF-induced PGCLCs (see lines 15-15, p.g. 11).
- Our co-culture system additionally follows the temporal order of germ cell development (late PGC>spermatogonia-like> early meiosis markers) (Fig. 4).

While triple-marker co-expression and the single-cell expression was not performed, we acknowledged this as a valuable future direction and highlighted it in the revised manuscript, see p.g. 20, lines 5-9.

3. The authors state that 4TF PGCLCs resemble *in vivo* PGCs, but details of this analysis are limited. It is unclear how homogeneous the PGCLC population is or whether subpopulations (e.g. partially differentiated or somatic-like cells) are present. The RNA-seq data should be examined for any residual somatic programs or incomplete reprogramming. Some methodological details could undermine interpretations. For instance, the Stella-eGFP reporter marks a germ cell gene, but Stella alone is not PGC-specific at all stages.

Response: We appreciate the reviewer's insightful comment. Transcriptomic analyses indicate that 4TF-induced PGCLCs closely resemble *in vivo* PGCs at E10.5–E11.5 and exhibit greater homogeneity compared to cytokine-induced PGCLCs (Fig. 2A). We revised our discussion to acknowledge that single-cell RNA sequencing will be valuable for characterizing subpopulations within our differentiation system, stating: "Although these findings indicate more advanced development in 4TF-induced PGCLCs, single-cell RNA sequencing will be required to provide detailed insights into their differentiation trajectories." on p.g. 19, lines 1-2.

4. The comparisons to "cytokine-induced" PGCLCs need clarification: are these data side-by-side from identical ESC lines and conditions? The baseline efficiency and quality of the cytokine method in the authors' hands is not shown, making it hard to judge the improvement.

Response: The Stella-eGFP ESC line, originally obtained from the Surani lab, typically yields ~40% Stella-eGFP⁺ cells by day 6 under cytokine-based induction conditions (8). We did not perform efficiency analyses in our lab side-by-side. We highlighted this efficiency comparison in the revised manuscript, stating: "...representing a two-fold increase in efficiency compared with the cytokine induction system using the same cell line" (p.g. 8, lines 4–5).

5. The finding that maternal HDACi exposure causes hypospermatogenesis is intriguing, but the evidence is tenuous. It is not shown whether this effect is direct on germ cells or secondary to other developmental defects (VPA is a known teratogen). HDAC activity is broadly required for cell differentiation; VPA and sodium butyrate are non-specific and can affect many lineages. The manuscript should distinguish general toxicity or differentiation block from germline-specific effects. Also, NCOR2 itself has many targets - is its effect in PGCLCs mediated only via HDAC recruitment, or through other co-repressors? The conclusion that NCOR2 is an "epigenetic fertility modifier" is plausible but not fully supported without deeper mechanistic insight.

It may help to use more selective HDAC inhibitors or genetic knockdown to pinpoint which HDAC isoforms are critical. For instance, test class I vs II-specific inhibitors, and knockdown Hdac3 (a class I) or Hdac4/5 (class IIa) separately in PGCLCs. Include cell viability and cell cycle assays to ensure that effects are not due to cytotoxicity. Conversely, test whether adding HDAC activators or overexpressing HDAC3/4 can enhance PGCLC formation. Additionally, include a non-germline differentiation control (e.g. differentiate ESCs to neurons or muscle under similar HDACi treatment) to show that the observed effect is germline-specific.

Response: We thank the reviewer for these thoughtful and detailed suggestions, which would undoubtedly provide deeper mechanistic insight into the role of NCOR2 and HDAC activity in PGCLC formation. We addressed some of these issues in the response to Reviewer 1's comments, point # 14. It is possible that there is some general toxicity, but in the ESC>PGCLC induction assays, what was affected was the efficiency of PGCLC formation (GFP⁺ vs GFP⁻ cells).

At this stage, we believe that our current data, which include both genetic (siRNA/CRISPRi) and pharmacological (HDACi) perturbations, provide a robust foundation for our main conclusions regarding the requirement for HDAC activity and NCOR2-mediated repression in germline fate enforcement. Our transcriptomic analyses already demonstrate upregulation of somatic genes and loss of PGCLC identity upon HDAC inhibition (Figs. EV4–EV5). Mochizuki et al. observed the upregulation of somatic genes and downregulation of PGC genes after *Ncor2* KD (9). These lines of evidence support a model in which these factors act to repress non-germline programs.

We fully acknowledge that more selective HDAC inhibition, isoform-specific knockdown, and non-germline differentiation controls would offer additional mechanistic clarity. However, these experiments are beyond the scope of the current study (they would take several months and funding we don't have), which was designed to establish the feasibility of high-throughput screening in PGCLCs and to identify epigenetic regulators that may point to environmental counterparts. We revised the Discussion (p.g. 21, lines 18-21) to explicitly note these limitations and to highlight the need for future studies addressing the specificity of HDAC isoforms, the role of other NCOR2 co-repressors, in addition to aforementioned edits we've made regarding potential for off-target effects.

6. The screen identified dozens of "hits" but only NCOR2 is discussed in detail. It's unclear how reproducible the screen was (technical replicates, library coverage, false positives). Were any other top hits followed up or validated individually? If NCOR2 is highlighted, why not other strong hits?

Response: The primary aim of our CRISPRi screen was to demonstrate the scalability and robustness of our 4TF PGCLC induction platform for high-throughput functional genomics. Our platform enables the generation of sufficient PGCLCs to perform CRISPRi screens with high coverage and statistical power. We maintained a minimum of 500-fold coverage per sgRNA (see "Lentiviral packaging and transduction" section in M&M). Such high coverage is critical for overcoming experimental noise and minimizing stochastic dropout, thereby reducing the risk of false positives and negatives. The full scale screen was done once, but only after we had ensured robust lentiviral library representation as described in the manuscript.

As a proof-of-concept for the 4TF platform's utility for high-throughput functional genomic screens, Ncor2 was selected for detailed follow-up because it ranked as a top hit, and is a known regulator of chromatin and germline fate. We summarized and discussed the additional top hits (see last paragraph of Suppl. Text). Validating and exploring additional top hits (as well as those from a CRISPRa screen we did not describe here) from the screen are in future plans, (pending funding) but fall outside the scope of the current study.

7. The authors could measure expression of somatic genes (e.g. Hoxb1, Hand1) or PGC markers in NCOR2-depleted cells, paralleling Mochizuki's findings. This is partly addressed in RNA-seq of Stella⁺ vs Stella⁻ cells under HDACi (Figs S15-S16). It may suffice to simply clarify in the main text that NCOR2 likely enforces germline fate by repressing somatic programs, as the siRNA and HDACi data imply.

Response: We thank the reviewer for this thoughtful and specific suggestion, which highlights the importance of directly linking NCOR2 function to the repression of somatic gene expression during PGCLC differentiation. We discussed the NCOR2 involvement in suppressing somatic programs in the Discussion section of the Supplementary Text, referencing the Mochizuki paper, but we did as the reviewer suggested and emphasize the point in the main text in our revision (see p.g. 15, lines 1-2).

8. To clarify the role of NCOR2 in PGCLC formation, it will be good to independently validate NCOR2's role by CRISPR knockout or RNAi in EpiLCs (rather than CRISPRi), and rescue the phenotype by overexpressing an HDAC-binding-deficient NCOR2 mutant to see if HDAC recruitment is necessary. Additionally, RNA-seq of NCOR2-knockdown PGCLCs could identify which germline and somatic genes are misregulated, helping distinguish primary regulatory effects (see pt#8).

Optional: If the authors wish to further explore chromatin-level mechanisms, ChIP-qPCR for histone acetylation (e.g. H3K27ac) at germline gene loci could help define how NCOR2-HDAC repression shapes gene accessibility. Additionally, while not essential for the core conclusion, assessing whether other top screen hits (~53 total) similarly influence germline-specific versus general differentiation pathways could provide a broader picture of epigenetic regulation in PGCLCs.

Response: We appreciate the reviewer's insightful comment guiding potential future investigations into these detailed mechanisms regarding NCOR2's role in PGCLC development. However, we feel that this is beyond the scope of the present report. The main focus of this paper is platform development and we used a CRISPRi screen to demonstrate the viability of our 4TF platform to conduct large scale functional screens. We are hopeful that funding will come through to follow up with more detailed studies such as those suggested (as well as some of the other hits from the screen), which would involve years of research.

9. While the 4TF PGCLCs display germ cell markers and epigenetic hallmarks of PGCs, it is unclear if they can complete germ cell development in vivo or in a more physiologically relevant assay. Claims of "spermatogonia-like" differentiation are based on in vitro morphology or marker expression, but the work lacks definitive evidence (e.g. progression through meiosis, haploid formation, or transplantation assays). Without demonstrating formation of mature germ cells or fertility restoration, the true developmental potential of these PGCLCs remains uncertain.

Response: We have acknowledged this limitation in the manuscript (p.g.20, lines 10-12). To characterize the induced PGCLCs, we employed widely accepted approaches (such as ref. (10)), including transcriptional profiling, analysis of epigenetic hallmarks, and reprogramming into pluripotent EGCs, as detailed in our manuscript. Importantly, our co-culture system offers functional support for germ cell identity of PGCLCs. At this stage, our study was primarily designed to establish the feasibility of high-efficiency PGCLC induction system and apply this platform for high-throughput screening. While in vitro spermatogenesis- particularly the successful completion of normal meiosis- remains a major challenge in the field, it falls outside the scope of the current study. We are considering establishing a testis injection platform via the efferent duct in neonatal (7- to 9-day-old) mice (11)(12) or seeking collaborations to enable this approach, although it is a highly specialized technique currently accessible in only a few laboratories worldwide.

Minor comments:

1. Line 205: The Results mention that "PGCLCs could be dedifferentiated to EGCs" when transferred to 2iLIF (Fig. S2). This is interesting and perhaps should be briefly discussed (even if just in Supp.), as it shows germline epigenetic resetting.

Response: Thanks for highlighting this point. We have added a couple of sentences to the discussion as follows on lines 2-5 of p.g. 19 in updated version: "Interestingly, 4TF-induced PGCLCs retain the capacity for conversion into pluripotent EGCs. These findings highlight the remarkable plasticity of the germline epigenome, though the differences in epigenetic reprogramming potential compared to endogenous late PGCs remain to be elucidated."

2. The alphanumeric figure labels are not intuitive and may be confusing; for example, in Fig. 2, 'D' appears after 'A' and is followed by 'B' and then 'C'.

Response: We adjusted the figures and revised accordingly.

3. The term "cost-effective" is used in the Abstract; authors could briefly quantify or justify this (e.g. "using small-molecule versus expensive cytokines").

Response: Revised to "yields abundant and highly enriched PGCLCs without costly recombinant cytokines.." Also, details on costs were addressed as mentioned in our response to point #1 by this reviewer.

4. The number of animals, dosage, and controls (e.g. unaffected maternal health) should be detailed.

Response: We provided these details in revised version, see section "Mouse treatment and reproductive phenotyping" in M&M.

References

1. Sato T, Katagiri K, Gohbara A, Inoue K, Ogonuki N, Ogura A, et al. In vitro production of functional sperm in cultured neonatal mouse testes. *Nature*. 2011 Mar 24;471(7339):504–7.
2. Enders GC, May JJ. Developmentally regulated expression of a mouse germ cell nuclear antigen examined from embryonic day 11 to adult in male and female mice. *Dev Biol*. 1994 Jun;163(2):331–40.
3. Carmell MA, Dokshin GA, Skaletsky H, Hu Y-C, van Wolfswinkel JC, Igarashi KJ, et al. A widely employed germ cell marker is an ancient disordered protein with reproductive functions in diverse eukaryotes. *eLife*. 2016 Oct 8;5.
4. Rodda DJ, Chew J-L, Lim L-H, Loh Y-H, Wang B, Ng H-H, et al. Transcriptional regulation of nanog by OCT4 and SOX2. *J Biol Chem*. 2005 Jul 1;280(26):24731–7.
5. Murakami K, Günesdogan U, Zyllicz JJ, Tang WWC, Sengupta R, Kobayashi T, et al. NANOG alone induces germ cells in primed epiblast in vitro by activation of enhancers. *Nature*. 2016 Jan 21;529(7586):403–7.
6. Zhang M, Leitch HG, Tang WWC, Festuccia N, Hall-Ponsele E, Nichols J, et al. Esrrb Complementation Rescues Development of Nanog-Null Germ Cells. *Cell Rep*. 2018 Jan 9;22(2):332–9.
7. Sasaki H, Matsui Y. Epigenetic events in mammalian germ-cell development: reprogramming and beyond. *Nat Rev Genet*. 2008 Feb;9(2):129–40.
8. Hackett JA, Huang Y, Günesdogan U, Gretarsson KA, Kobayashi T, Surani MA. Tracing the transitions from pluripotency to germ cell fate with CRISPR screening. *Nat Commun*. 2018 Oct 16;9(1):4292.
9. Mochizuki K, Hayashi Y, Sekinaka T, Otsuka K, Ito-Matsuoka Y, Kobayashi H, et al. Repression of somatic genes by selective recruitment of HDAC3 by BLIMP1 is essential for mouse primordial germ cell fate determination. *Cell Rep*. 2018 Sep 4;24(10):2682–2693.e6.

10. Yu L, Wei Y, Sun H-X, Mahdi AK, Pinzon Arteaga CA, Sakurai M, et al. Derivation of intermediate pluripotent stem cells amenable to primordial germ cell specification. *Cell Stem Cell*. 2021 Mar 4;28(3):550-567.e12.
11. Nakaki F, Hayashi K, Ohta H, Kurimoto K, Yabuta Y, Saitou M. Induction of mouse germ-cell fate by transcription factors in vitro. *Nature*. 2013 Sep 12;501(7466):222–6.
12. Hayashi K, Ohta H, Kurimoto K, Aramaki S, Saitou M. Reconstitution of the mouse germ cell specification pathway in culture by pluripotent stem cells. *Cell*. 2011 Aug 19;146(4):519–32. Sciwheel inserting bibliography...

Dear Prof. Schimenti,

Thank you for the submission of your revised manuscript to our editorial offices. I have now received the report from one of the two referees that I asked to re-evaluate the study, you will find below. Referee #2 was completely unresponsive to my invitations to re-assess the manuscript. However, going through your detailed p-b-p-response and the revised manuscript, I consider his/her points as adequately addressed. Referee #1 now fully supports the publication of your study in EMBO reports.

Before we can proceed with formal acceptance, I have the following editorial requests I ask you to address in a final revised manuscript:

- Please provide a final comprehensive title with not more than 100 characters (including spaces).
- Please provide the abstract written in present tense throughout.
- Please add up to five keywords to the manuscript and order the sections like this, using only these names:
Title page - Abstract - Keywords - Introduction - Results - Discussion - Methods - Data availability section - Acknowledgements (please put here all the funding information) - Disclosure and Competing Interests Statement - References - Figure legends - Expanded View Figure legends
- There is a file named 'Supplementary Text' uploaded and called out. Please incorporate this text into the main manuscript text and add the references to the main references. We do not allow supplementary text or methods.
- Please name file 'Supplementary Figures' Appendix. The Appendix should have page numbers and needs to include a table of content on the first page (with page numbers). There is no need to repeat the author information, though. It is sufficient to state 'Appendix for ...' followed by the final title, with the TOC below. Please follow the nomenclature 'Appendix Figure Sx' throughout the text (for the callouts) and also label the figures in the Appendix file according to this nomenclature.
- There are presently 21 Supplementary Figures mentioned ('Supplementary Figures 1~21'), but the file only contains 16. Please check.
- Please add scale bars of similar style and thickness to all microscopic images (main, EV and Appendix images), using clearly visible black or white bars (depending on the background). Please place these in the lower right corner of the images themselves. Please do not write on or near the bars in the image but define the size in the respective figure legend. Presently, scale bars are missing from most of the images.
- Please check again that the number "n" for how many independent experiments were performed, their nature (biological versus technical replicates), the bars and error bars (e.g. SEM, SD) and the test used to calculate p-values is indicated in the respective figure legends. Please also check that all the p-values are explained in the legend, and that these fit to those shown in the figure. Please provide statistical testing where applicable. Please avoid the phrase 'independent experiment' but clearly state if these were biological or technical replicates. Please also indicate (e.g. with n.s.) if testing was performed, but the differences are not significant. In case n=2, please show the data as separate datapoints without error bars and statistics. See also:
<http://www.embopress.org/page/journal/14693178/authorguide#statisticalanalysis>
- If n<5, please show single datapoints for diagrams. Moreover:
 - Please note that the exact p values are not provided in the legends of figures 5C, EV3 B, F, G, I
 - Please indicate the statistical test used for data analysis in the legends of figures EV3 A, D
 - Please note that the box plots need to be defined in terms of minima, maxima, centre, bounds of box and whiskers, and percentile in the legends of figures 1F, 5C
 - Please note that information related to n is missing in the legends of figures 1G, 5C, E, G, H; 7D, EV3 B, D, F, G, I
 - Please note that the dotted borders are not defined in the legend of figure 4B. This needs to be rectified.
- Please separate the different tables provided as 'Dataset 1'. Please upload Tables S1, S2, S3 and S4 as separate dataset files named Dataset EV1, Dataset EV2, Dataset EV3 and Dataset EV4. Please add a legend for each on the first TAB of the respective excel file and change the callouts or add separate callouts for these.
- Please add the primer and antibody information provided in tables S5 and S6 directly to the Reagents & Tools table. Please also remove the Reagents & Tools Table from the main manuscript file and upload it as separate file using the provided template:

<https://www.embopress.org/page/journal/14693178/authorguide#structuredmethods>

Finally, please add callouts to the table to the methods section were appropriate.

- Thanks for providing the source data for the microscopic image. However, please also provide numerical source data were appropriate, e.g. for the panels 1A, 1F, 1G, 5B, %C, 5E, 5H, 5G, 6C, 6D and 7D. Please add these to the SD folders for the respective figures.

In addition, I would need from you uploaded separately:

Best,

Referee #1:

The authors were responsive to the previous critiques, and the manuscript has been substantially improved.

All editorial and formatting issues were resolved by the authors.

Prof. John Schimenti
Cornell University
Biomedical Sciences
T9014A Vet Research Tower
T9014A Vet Research Tower
Ithaca, NY 14853
United States

Dear Prof. Schimenti,

I am very pleased to accept your manuscript for publication in the next available issue of EMBO reports. Thank you for your contribution to our journal.

Yours sincerely,
